# Immunogenicity and efficacy of vaccine boosters against SARS-CoV-2 Omicron subvariant BA.5 in male Syrian hamsters

Rafael R. G. Machado [1,2], Jordyn L. Walker[1,3], Dionna Scharton [1,3], Grace H. Rafael[1], Brooke M. Mitchell[1,3], Rachel A. Reyna [1,4], William M. de Souza[1], Jianying Liu[1], David H. Walker[4,5,6], Jessica A. Plante [1,3], Kenneth S. Plante [1,3,6] ✉ & Scott C. Weaver [1,3,5,6] ✉

The SARS-CoV-2 Omicron subvariant BA.5 rapidly spread worldwide and replaced BA.1/BA.2 in many countries, becoming globally dominant. BA.5 has unique amino acid substitutions in the spike protein that both mediate immune escape from neutralizing antibodies produced by immunizations and increase ACE2 receptor binding affinity. In a comprehensive, long-term (up to 9 months post primary vaccination), experimental vaccination study using male Syrian hamsters, we evaluate neutralizing antibody responses and efficacy against BA.5 challenge after primary vaccination with Ad26.COV2.S (Janssen) or BNT162b2 (Pfizer/BioNTech) followed by a homologous or heterologous booster with mRNA-1273 (Moderna) or NVX-CoV2373 (Novavax). Notably, one high or low dose of Ad26.COV2.S provides more durable immunity than two primary doses of BNT162b2, and the NVX-CoV2373 booster provides the strongest augmentation of immunity, reduction in BA.5 viral replication, and disease. Our data demonstrate the immunogenicity and efficacy of different prime/boost vaccine regimens against BA.5 infection in an immune-competent model and provide new insights regarding COVID-19 vaccine strategies.

The COVID-19 pandemic prompted an unprecedented effort to rapidly develop and deploy with billions of doses administered worldwide[1]. Several different platforms have been used to develop the currently approved SARS-CoV-2 vaccines[2,3], including replication-defective viral vectors (Ad26.COV2.S produced by Janssen and AZD1222 produced by AstraZeneca), mRNA (BNT162b2 produced by Pfizer/BioNTech and mRNA-1273 produced by Moderna), and protein subunits (NVX-CoV2373 produced by Novavax), among others. These vaccines have reduced the numbers of COVID-19-related infections, hospitalizations,

and deaths[4,5]. However, the emergence of more transmissible SARS-CoV-2 variants of concern (VOCs) has caused cyclical infection waves, including Alpha (B.1.1.7), Delta (B.1.617) and more recently, Omicron (B.1.1.529)[6]. These VOCs include multiple amino acid substitutions in the spike protein, the primary target of neutralizing antibodies, raising concerns regarding the efficacy of global vaccination strategies. In response, several countries have adopted the administration of booster doses, following WHO recommendations, to augment immunity and protection[7].

[1]Department of Microbiology and Immunology, University of Texas Medical Branch, Galveston, TX 77555, USA. [2]Department of Microbiology, Institute of Biomedical Sciences, University of Sao Paulo, Sao Paulo, SP 05508000, Brazil. [3]World Reference Center for Emerging Viruses and Arboviruses, University of Texas Medical Branch, Galveston, TX 77555, USA. [4]Department of Pathology, University of Texas Medical Branch, Galveston, TX 77555, USA. [5]Center for Biodefense and Emerging Infectious Diseases, University of Texas Medical Branch, Galveston, TX 77555, USA. [6]Institute for Human Infections and Immunity, University of Texas Medical Branch, Galveston, TX 77555, USA. ✉e-mail: ksplante@utmb.edu; sweaver@utmb.edu

BA.5 is a distinct Omicron lineage, with an estimated origin in early January 2022[8,9], and characterized by additional spike mutations 69-70del, L452R, F486V, and an ancestral amino acid at position Q493, compared to BA.2[8]. This subvariant has been correlated with increasing rates of reinfections, breakthrough cases and hospitalizations[10–13]. To optimally protect populations against BA.5, experimental studies with animal models may inform vaccination strategies.

Here, we evaluate the neutralizing antibody response and efficacy of homologous and two heterologous boosters (mRNA-1273 or NVX-CoV2373) following primary vaccination with either BNT162b2 or Ad26.COV2.S in male Syrian hamsters. We show the kinetics of the humoral response to the Ad26.COV2.S and BNT162b2 COVID-19 vaccines in Syrian hamsters through 9 months, and we evaluate the effect of several boosters. The Ad26.COV2.S at either low or high doses provides relatively stable neutralizing antibody levels, and the heterologous boost with NVX-CoV2373 vaccine results in a substantial increase in humoral immune responses and the most robust protection against Omicron BA.5 infection. These data contribute to our understanding of heterologous and homologous COVID-19 vaccine strategies and can contribute to recommendations for vaccination of humans.

## Results

### Immunogenicity of one-dose of Ad26.COV2.S and two-doses of BNT162b2 in Syrian hamsters

Syrian hamsters (HsdHan: AURA strain; 4–5 wk-old) were immunized with either a single dose of $10^{10}$ or $10^8$ viral particles (vp) of Ad26.COV2.S ($n = 50$ and 49, respectively) or with two doses of 5 μg of BNT162b2 in over 4-wk interval ($n = 50$). A sham-vaccinated control group ($n = 56$) received an injection of PBS at week 0 (Fig. 1a). Neutralizing (nAbs) and binding (bAbs) antibody responses against the SARS-CoV-2 USA-WA1/2020 ancestral strain (same sequence used for the spike proteins in the vaccines) were measured up to 252 days after the primary immunization by plaque reduction neutralization tests (PRNT) and by enzyme-linked immunosorbent assay (ELISA), respectively.

A peak of neutralizing antibodies was detected 56 days after initial vaccination with $10^{10}$ or $10^8$ vp of Ad26.COV2.S (geometric mean titers (GMTs): 368 and 139, respectively) (Fig. 1b, c) and 28 days after immunization with two 5 μg doses of BNT162b2 (GMT: 65; Fig. 1d). At the same time-points, a peak of binding antibodies was observed (Fig. 1e–g). At day 56 post-immunization, animals that received $10^{10}$ vp Ad26.COV2.S had consistently higher neutralizing antibody levels compared to animals immunized with either $10^8$ vp Ad26.COV2.S (1.6-fold and 2.6-fold, respectively; $p < 0.0001$ for both, $t$-test) or with BNT162b2 (5.8-fold and 9.9-fold, respectively; $p < 0.0001$ for both, $t$-test). Similar differences in the antibody response levels were maintained throughout the entirety of the study. At 252 days post-vaccination (dpv), the last pre-challenge time-point analyzed, animals that received $10^{10}$ vp Ad26.COV2.S vaccine had 2.6-fold higher binding ($p < 0.0001$, $t$-test) and 1.8-fold higher neutralizing ($p = 0.0005$, $t$-test) antibody titers than animals that received $10^8$ vp Ad26.COV2.S. The $10^{10}$ vp dose of Ad26COV2.S also maintained its advantage over primary vaccination with BNT162b2 at 252 dpv, with 14.0-fold higher binding ($p < 0.0001$, $t$-test) and 9.3-fold higher neutralizing antibody titers ($p < 0.0001$, $t$-test) relative to animals immunized with $10^8$ vp Ad26.COV2.S or with 5 μg BNT162b2, respectively (Fig. 1b–e). Sham-vaccinated animals had no detectable binding and neutralizing antibodies against USA-WA1/2020 during the study period (Supplementary Fig. 1).

Clear differences in the antibody kinetic profiles among primary vaccination groups were observed, with a faster and greater decrease in antibody levels after two doses of 5 μg BNT162b2 compared to the more durable neutralizing antibody titers after vaccination with either $10^8$ or $10^{10}$ vp of the Ad26.COV2.S vaccine (Fig. 1b–j).

### Neutralizing antibody responses after homologous and heterologous boosters

Six months (day 168) after the initial immunization, hamsters received a boost immunization (Supplementary Fig. 2), either a homologous booster with $10^8$ vp of Ad26.COV2.S or 0.5 μg BNT162b2, or a heterologous booster with either 0.5 μg mRNA-1273 or 1 μg NVX-CoV2373 (Supplementary Fig. 2). An age-matched sham-boosted (PBS) group was included for comparison. Neutralizing antibody activity against BA.5 was evaluated in samples from day 168 (immediately pre-boost), 196 (1-month post-boost) and 252 (3 months post-boost and immediately pre-challenge) post-vaccination. Pre-boost nAbs titers against BA.5 were statistically higher in animals primed with the $10^{10}$ vp dose of Ad26.COV2.S (GMT: 29) compared to those primed with either the $10^8$ vp dose of Ad26.COV2.S (GMT: 16, $p = 0.0002$) or with BNT162b2 (GMT: 11, $p < 0.0001$) (Supplementary Fig. 3) No significant difference was observed between $10^8$ vp Ad26.COV2.S and BNT162b2 ($p = 0.3005$; Supplementary Fig. 3).

The neutralizing activity against BA.5 was significantly different between the pre- and 1 month post-boost serum samples in the groups boosted with NVX-CoV2373, with a 2.6-, 3.8- and 2.5-fold increase in the groups that received primary vaccination with $10^{10}$ vp of Ad26.COV2.S ($n = 22$, GMT:80, $p = 0.006$), $10^8$ vp of Ad26.COV2.S ($n = 22$, GMT:50, $p = 0.013$) or 5 μg BNT162b2 ($n = 22$, GMT:32, $p = 0.024$), respectively (Fig. 2a–c, third column of graphs). Furthermore, 3 months post-boost, the GMTs remained stables and were not significantly different from 1 month post-boost ($p > 0.05$), while maintaining slightly higher antibody titers when compared with the pre-boost levels (Fig. 2a–c). In the hamsters that only received a prime vaccination with either a low or high dose of Ad26.COV2.S, and no boost, a larger increase in nAb titers was observed in animals boosted with NVX-CoV2373 (2.4- and 1.9-fold, $p > 0.05$) compared to mRNA-1273 (1.3- and 1.0-fold, $p > 0.05$) or Ad26.COV2.S (0.9- and 1.1-fold, $p > 0.05$) 1 month post-boost (Fig. 2d, e). A similar pattern was observed 3 months post-boost (Fig. 2d, e). One and three-months post-boost, animals primed with BNT162b2 and boosted with NVX-CoV2373 also showed the highest -fold difference (3.2- and 3.4-fold, $p < 0.006$) when compared to other boosting vaccines such as mRNA-1273 (1.7- and 1.6-fold, $p > 0.05$) and BNT162b2 (1.5- and 1.2-fold, $p > 0.05$)-boosted animals, using sham-boosted animals as a reference (Fig. 2f).

After 3 months post-boost, a higher proportion of animals boosted with NVX-CoV2373 and primed with either $10^8$ vp of Ad26.COV2.S or 5 μg of BNT162b2 showed measurable $FRNT_{50}$ titers against BA.5 (LOD = 20); 54% ($n = 7/13$) of the animals primed with $10^8$ vp of Ad26.COV2.S, and sham-boosted, developed $FRNT_{50}$ titers above the LOD (Supplementary Fig. 4 and Supplementary Table 1). Hamsters boosted with mRNA-1273, NVX-CoV2373 and Ad26.COV2.S developed a measurable nAb response at rates of 50% ($n = 6/12$), 75% ($n = 9/12$) and 58% ($n = 7/12$), respectively (Supplementary Table 1). Similar data were observed in the animals primed with 5 μg of BNT162b2, where no unboosted hamsters developed a measurable nAb response compared to 33% ($n = 4/12$), 67% ($n = 8/12$), and 25% ($n = 3/12$) of those boosted with mRNA-1273, NVX-CoV2373, or BNT162b2, respectively (Supplementary Fig. 4 and Supplementary Table 1).

### Effects of homologous and heterologous booster on protection against Omicron BA.5 challenge

Three days after pre-challenge bleeds, hamsters were infected via the intranasal route with $10^4$ PFU of Omicron BA.5. Infectious viral titers at 2 and 4 days post-infection (dpi) were measured in nasal washes, trachea, and lungs. Daily weights and signs of disease were recorded up to 4 dpi (Fig. 3a). In sham-vaccinated hamsters, Omicron BA.5 produced significant weight loss compared to mock-challenged controls at 3 ($p = 0.0027$; Fig. 3B) and 4 dpi ($p = 0.0009$, Fig. 3b). Compared to the sham-vaccinated group, the group primed with $10^{10}$ vp of Ad26.COV2.S (Fig. 3b, left panel), followed either with the heterologous (mRNA-1273

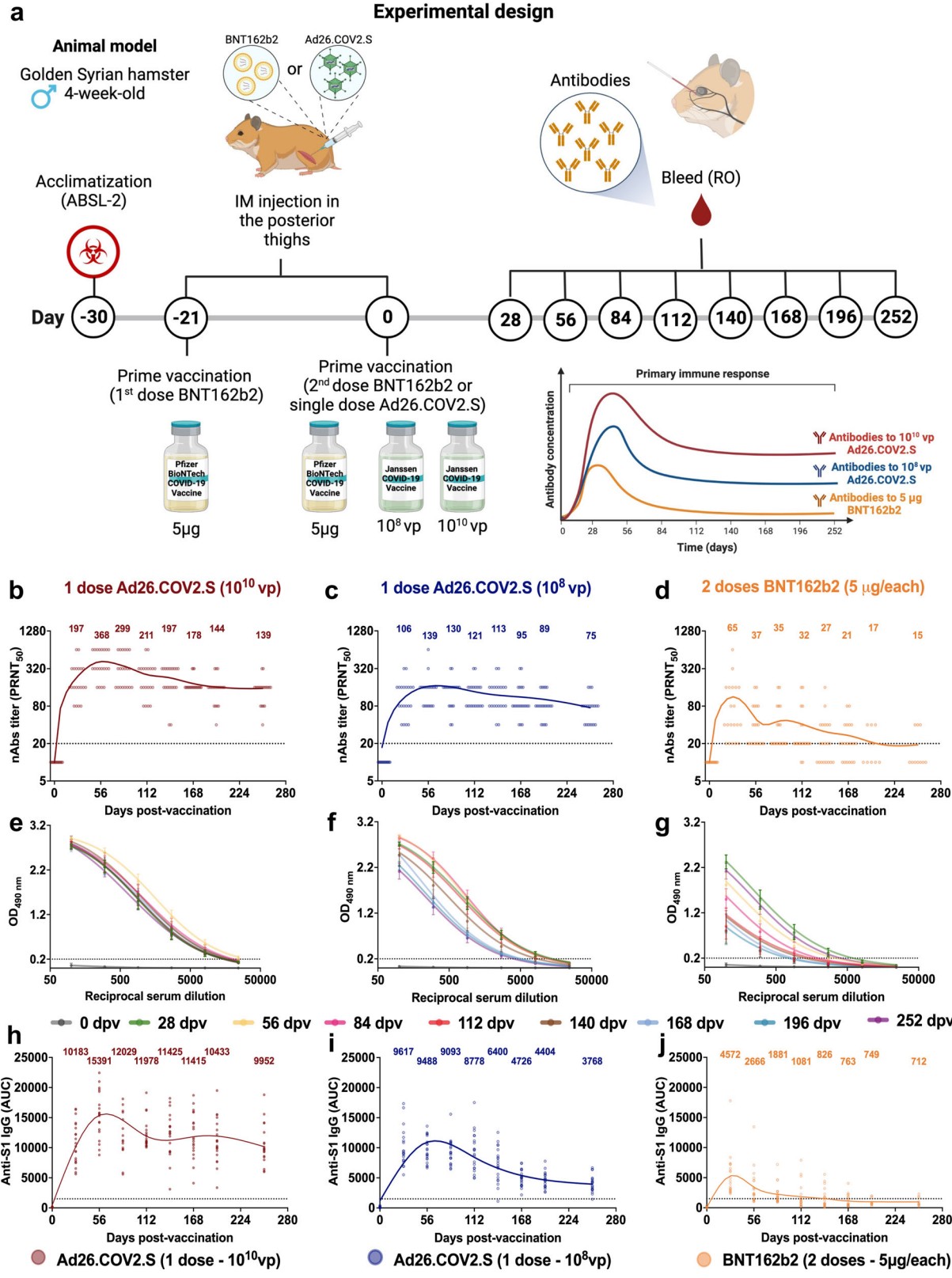

and NVX-CoV2373) or homologous (Ad26.COV2.S) boosters showed significantly less weight loss at 3 dpi ($p = 0.016$, $p = 0.006$ and $p = 0.002$, respectively) and 4 dpi ($p = 0.006$, $p = 0.002$ and $p = 0.008$, respectively) after BA.5 challenge. Similar prevention of weight loss was observed with the heterologous boosters in the group primed with $10^8$ vp of Ad26.COV2.S (Fig. 3b, middle panel) at 3 (mRNA-1273, $p = 0.0435$; NVX-CoV2373, $p = 0.0121$) and 4 dpi (mRNA-1273,

$p = 0.0236$; NVX-CoV2373, $p = 0.0027$). No statistical significance was observed in these groups when compared to the mock group at 3 and 4 dpi ($p > 0.05$). The homologous booster after $10^8$ vp of Ad26.COV2.S showed no difference at 3 dpi ($p = 0.7637$) and only a slight reduction in weight loss at 4 dpi compared to sham-vaccinated ($p = 0.0177$).

In animals primed with two doses of BNT162b2 (5 µg/each), the homologous booster provided significant protection against

**Fig. 1 | Antibody response kinetics of one dose Ad26.COV2 ($10^8$ or $10^{10}$ vp) and two doses BNT162b2 (5 μg/each) vaccines in Syrian hamsters.** Four-to-five-week-old male Syrian hamsters were immunized with one $10^8$ or $10^{10}$ vp dose of the Ad26.COV2.S vaccine ($n = 49$ and 50, respectively), two 5 μg doses of the BNT162b2 vaccine ($n = 50$), or sham-vaccinated (PBS; $n = 59$) and followed through 252 days post-vaccination (dpv). **a** Experimental design, showing the immunizations and blood collections. **b–d** Serum neutralizing antibody responses through 252 dpv after (**b**) $10^{10}$ vp of Ad26.COV2.S, (**c**) $10^8$ vp of Ad26.COV2.S, or (**d**) two 5 μg doses of BNT162b2 as determined by plaque reduction neutralization tests (PRNT) based on a 50% or greater reduction in plaque counts (PRNT$_{50}$) against the USA_WA1/2020 strain ($n = 20$/per group). **e–g** Serum binding antibody responses through 252 dpv after (**h**) $10^{10}$ vp of Ad26.COV2.S, (**i**) $10^8$ vp of Ad26.COV2.S, or (**j**) two 5 μg doses of BNT162b2 as determined by optical density at 490 nm (OD$_{490\ nm}$) in an ELISA

measuring anti-S1 (USA_WA1/2020) IgG ($n = 20$/per group). **h–j** Serum binding antibody responses through 252 dpv after (**e**) $10^{10}$ vp of Ad26.COV2.S, (**f**) $10^8$ vp of Ad26.COV2.S, or (**g**) two 5 μg doses of BNT162b2 as determined by area under the curve (AUC) kinetics in an ELISA measuring anti-S1 (USA_WA1/2020) IgG ($n = 20$/per group). For panels **b–d** and **h–j**, points represent individual subjects, fitted curves reflect the Fit Spline algorithm, dotted lines represent the lower limit of detection, and numbers represent the geometric mean titers (GMTs) at a given timepoint. For panels **e–g**, data are presented as GMTs, error bars represent the 95% confidence interval, and fitted curves reflect the Sigmoidal 4-Parameter Logistic (4PL) regression algorithm. Panel **a** was created with BioRender.com. Panels **b–j** were generated using GrapPad Prism v9.4 software. See Source Data for complete data and Supplementary Fig. 2 for complete vaccination regimens scheme.

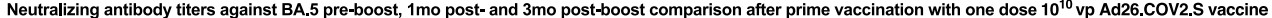

**Neutralizing antibody titers against BA.5 pre-boost, 1mo post- and 3mo post-boost comparison after prime vaccination with one dose $10^{10}$ vp Ad26.COV2.S vaccine**

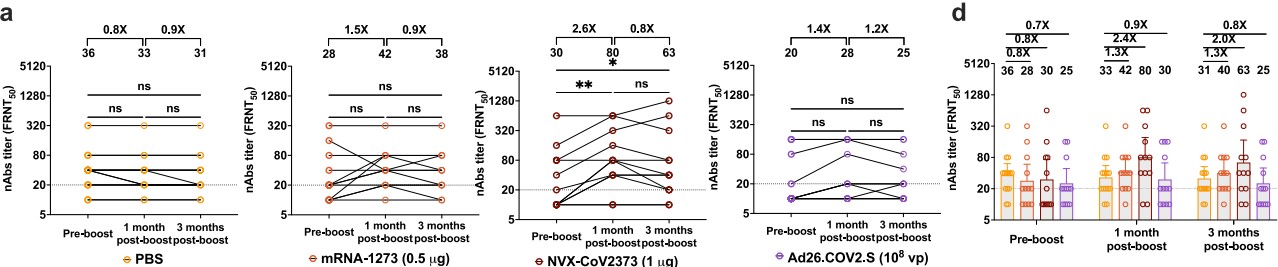

**Neutralizing antibody titers against BA.5 pre-boost, 1mo post- and 3mo post-boost comparison after prime vaccination with one dose $10^8$ vp Ad26.COV2.S vaccine**

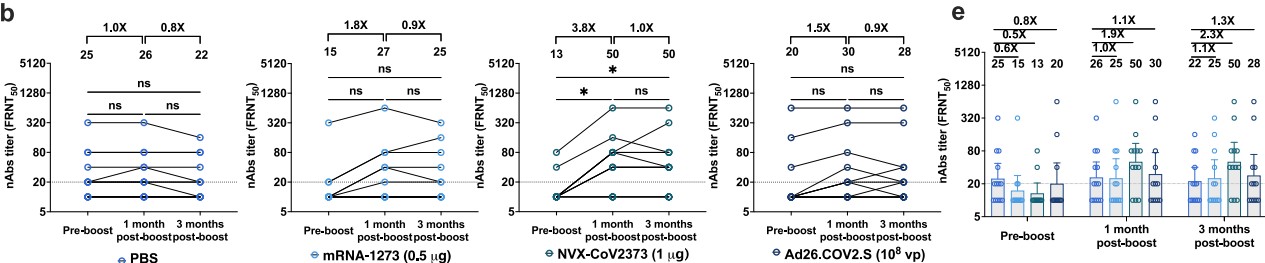

**Neutralizing antibody titers against BA.5 pre-boost, 1mo post- and 3mo post-boost comparison after prime vaccination with two doses BNT162b2 (5 μg/each)**

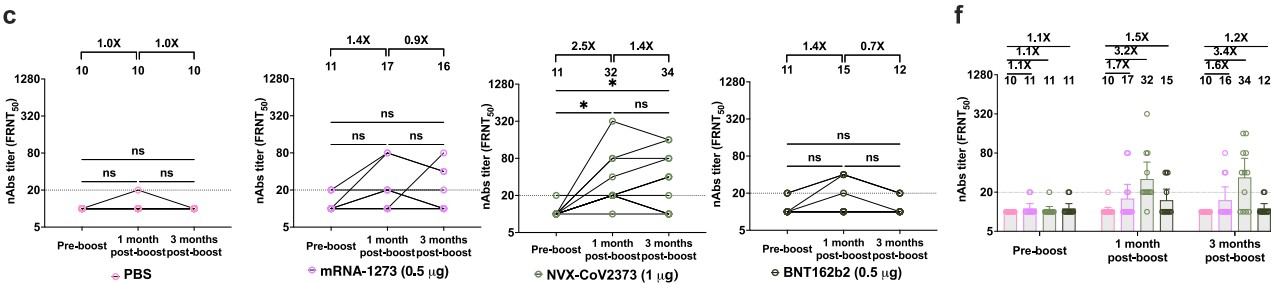

**Fig. 2 | Booster dose of NVX-CoV2373 enhanced neutralizing antibody responses against BA.5 in hamsters primarily vaccinated with Ad26.COV2.S or BNT162b2.** Four-to-five-week-old male hamsters were immunized either with one dose ($10^{10}$ or $10^8$ vp) of Ad26.COV2.S or two doses of BNT162b2 (5 μg per dose). After 6 months (day 168), animals were boosted with heterologous vaccines: 0.5 μg mRNA-1273 ($n = 12$ per group) or 1 μg rS/15 μg Matrix-M NVX-CoV2373 ($n = 12$ per group), or with the homologous vaccines: $10^8$ vp Ad26.COV2.S ($n = 12$ per group) or 0.5 μg BNT162b2 ($n = 12$). A sham-boosted group (PBS) was included for comparison ($n = 13$ for $10^8$ vp of Ad26.COV2.S or $n = 14$ for $10^{10}$ vp of Ad26.COV2.S and BNT162b2 groups). Blood was collected immediately prior to boosting as well as at 1 and 3 months post-boost (days 196 and 252). Neutralizing antibody titers against Omicron BA.5 were determined by foci reduction neutralization assay (FRNT), with the FRNT$_{50}$ values defined as the maximum serum dilution to neutralize at least 50% of infectious virus. **a–c** Paired analysis of pre-boost, 1 month post-boost, and

3 months post-boost serum neutralizing titers against Omicron BA.5 from hamsters that received as primary immunizations (**a**) $10^{10}$ vp Ad26.COV2.S, (**b**) $10^8$ vp Ad26.COV2.S, or (**c**) two 5 μg doses of BNT162b2. **d–f** Grouped analysis of serum neutralizing antibody responses of samples collected pre-boost, 1 month post-boost and 3 months post-boost against BA.5 in hamsters that received as primary immunizations (**d**) $10^{10}$ vp Ad26.COV2.S, (**e**) $10^8$ vp Ad26.COV2.S, or (**f**) two 5 μg doses of BNT162b2. Colors designate experimental cohorts, data points represent individual subjects, bar heights represent the geometric mean titer (GMT), error bars represent the 95% confidence interval, and dotted lines represent the lower limit of detection. GMTs and fold-change values are noted above the respective cohorts. Groups were compared by non-parametric ANOVA (Friedman's test) with Dunn's multiple comparison post-test (ns = not significant, *$p < 0.05$, **$p < 0.01$, ***$p < 0.001$, ****$p < 0.0001$). All panels were generated using GrapPad Prism v9.4 software. See Source Data for complete data.

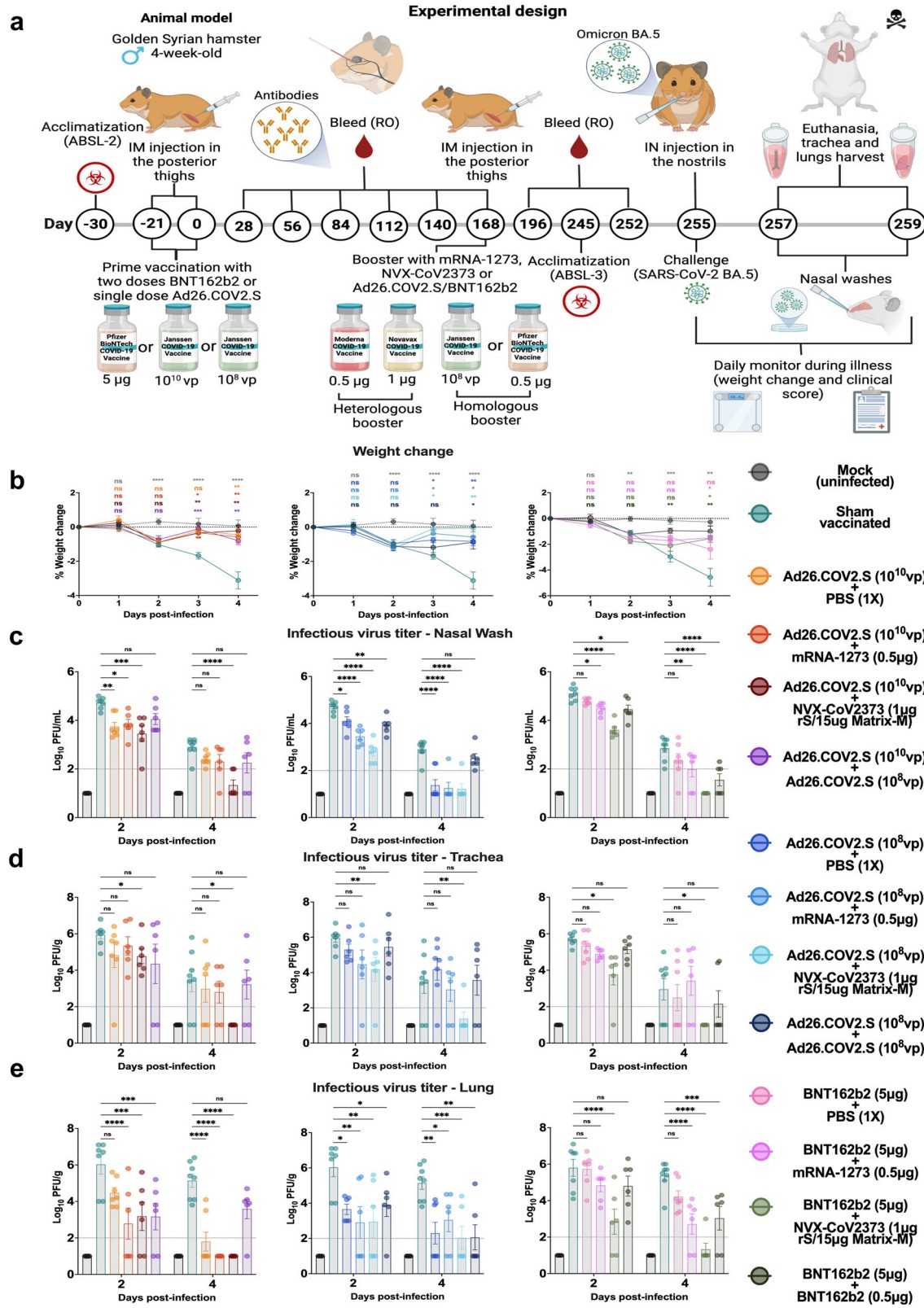

weight loss on 3 ($p = 0.0066$) and 4 dpi ($p = 0.004$) following challenge with BA.5. Both heterologous boosters showed similar results, differing significantly from the sham-vaccinated group only at 4 dpi (mRNA-1273, $p = 0.0109$; NVX-CoV2373, $p = 0.0116$). No significant differences in weight loss were observed on days 1 and 2 post-infection in any groups infected with BA.5. Compared to the sham-vaccinated animals, the sham-boosted group showed better

protection from weight loss on days 2–4 post-infection (Fig. 3b), indicating that the primary vaccination series did provide some protection against post-challenge weight loss. Furthermore, as Omicron challenge failed to produce any signs of disease in the hamsters, such as lethargy or ruffled fur, we could not use this metric to evaluate the prime/boost combinations against BA.5 (Supplementary Fig. 5).

**Fig. 3 | Booster dose of NVX-CoV2373 enhances protection against BA.5 infection of Golden hamsters after primary vaccination with one dose of Ad26.COV2.S or two doses of BNT162b2.** Four-to-five-week-old male hamsters were immunized either with one dose ($10^{10}$ or $10^8$ vp) of Ad26.COV2.S or two doses of BNT162b2 (5 μg per dose) vaccines. After 6 months (day 168), animals were boosted with heterologous vaccines: (0.5 μg mRNA-1273 or 1 μg rS/15 μg Matrix-M NVX-CoV2373), or with the homologous vaccines ($10^8$ vp Ad26.COV2.S or 0.5 μg BNT162b2). Sham-vaccinated and sham-boosted groups were included for comparison. At 3 months post-boost, hamsters were challenged with Omicron BA.5 or mock (PBS). **a** Schematic diagram of immunizations, blood draws, and virus challenge. **b** Change in body weight ($n = 129$ at 0–2 dpi, $n = 80$ at 3-4 dpi). **c–e** Infectious virus titers at 2 and 4 dpi in (**c**) nasal washes, (**d**) trachea, and (**e**) lungs after BA.5 or mock challenge. Data are organized based on the primary vaccination with $10^{10}$ vp of Ad26.COV2.S (left), $10^8$ bp of Ad26.COVV2.S (center), or two 5 μg doses of BNT162b2 (right) vaccines; sham data for the Ad26.COVV2.S groups were identical but different from the BNT162b2 group. Colors designate experimental cohorts ($n = 6$–8 subjects per group). In panel B, data points represent the sample means and error bars represent the standard deviations. In panels **c–e**, data points represent individual subjects, bar heights represent the sample means, error bars represent the standard deviations, and dotted lines represent the lower limit of detection. Groups were compared by one-way ANOVA with Tukey's post-test (ns = not significant, $*p < 0.05$, $**p < 0.01$, $***p < 0.001$, $****p < 0.0001$). Panel A was created with BioRender.com. Panels B–E were generated using GraphPad Prism v9.4 software. See Source Data for complete data.

Comparing infectious virus titers in the nasal washes, trachea and lungs, hamsters boosted with NVX-CoV2373 showed the highest level of protection, independent of the primary vaccination (Fig. 3c–e). Hamsters primed with the high dose of Ad26.COV2.S and boosted with NVX-CoV2373 showed 9-, 3- and 77- fold reductions at 2 dpi and 32-, 86,000- and 690,000-fold reductions at 4 dpi in the nasal washes, trachea and lungs, respectively (Fig. 3c–e and Supplementary Table 2), compared to sham-vaccinated hamsters. In addition, animals primed with either the high or low dose of Ad26.COV2.S and boosted with mRNA-1273 or Ad26.COV2.S vaccines, showed similar levels of protection, except on day 4 post-infection, when higher -fold reductions in viral load, mainly in the lower respiratory tract, were observed in animals boosted with mRNA-1273 (Supplementary Table 2). Surprisingly, hamsters that received a single dose of either $10^{10}$ or $10^8$ vp of Ad26.COV2.S vaccine without boosting also showed relatively strong protection against infectious viral loads with an -10-fold reduction in PFU per gram/ml in the samples measured, similar to that of the homologous boosted group (Fig. 3b–d, left and middle panels; Supplementary Table 2).

Hamsters primed with two 5 μg doses of BNT162b2 and boosted with either mRNA-1273 or BNT162b2 showed equivalent protection against BA.5 infection at 4 dpi (Fig. 3b–d, right panels). However, the heterologous booster with mRNA-1273 provided significantly better protection at the early time-point (2 dpi), with 4-, 69- and 20-fold reductions in viral loads in nasal washes, trachea, and lungs, respectively, compared to 3-, 25- and 7-fold reductions in the group boosted with the homologous vaccine (Supplementary Table 2). No significant protection was observed in sham-boosted animals primed with BNT162b2, compared with the unvaccinated controls (Fig. 3b–d, right panels), highlighting the importance of an additional boost to protect against this BA.5 variant.

We also performed histopathological evaluation of lung tissues from immunized and sham-vaccinated hamsters challenged with BA.5 or mock-challenged animals. Lung sections obtained at 4 dpi from sham-vaccinated animals showed pulmonary lesions with moderate to severe interstitial pneumonia characterized by hemorrhages, perivasculitis, perivascular edema, arterial endothelial leukocytic margination, peribronchiolitis, vasculitis and bronchiolitis (Fig. 4a). In contrast, hamsters primed with $10^8$ or $10^{10}$ vp of Ad26.COV2.S, with or without boosting, showed only mild peribronchitis, minimal interstitial pneumonia (few small foci) and mild perivasculitis. Few apparent differences were observed among groups receiving different vaccines or boosters (Fig. 4b and Supplementary Fig. 6). Animals primed with two doses of BNT162b2 (5 μg per dose) and boosted with either mRNA-1273 or NVX-CoV2373 showed a reduction in severity of pulmonary lesions, with only few small foci of peribronchiolar and perivascular mononuclear cells (Fig. 4b), as showed in the histopathological score (Supplementary Fig. 6). Sham-boosted and homologous-boosted hamsters had similar pathology scores (Supplementary Fig. 6) and pulmonary lesions to sham-vaccinated animals, with moderate interstitial pneumonia, peribronchiolitis, arterial endothelial leukocytic margination, bronchiolitis, peribronchitis and perivasculitis (Fig. 4a,b).

## Correlation between neutralizing antibody titers and protection against BA.5

Serum neutralizing antibody titers detected at 3 months post-boost (pre-challenge) showed an expected inverse correlation with viral loads in the nasal washes, trachea, and lungs at 2 dpi (Supplementary Fig. 7) and 4 dpi (Fig. 5). The strongest correlation was observed in the lungs of BA.5-challenged animals at 4 dpi, as exhibited by the animals vaccinated with $10^{10}$ vp of Ad26.COV2.S (Spearman's rank correlation coefficient $r = -0.86$, $p < 0.0001$; Fig. 5c), $10^8$ vp Ad26.COV2.S (Spearman's rank correlation coefficient $r = -0.80$, $p < 0.0001$; Fig. 5f) and two 5 μg doses BNT162b2 (Spearman's rank correlation coefficient $r = -0.77$, $p < 0.0001$; Fig. 5i). A detectable neutralizing titer ($FRNT_{50} \geq 20$) appeared to completely prevent infection of the lungs at 4 dpi, with predictive values of 91% for negative (95%CI: 80–96%) and 98% for positive loads (95%CI: 87–100%), respectively (Supplementary Table 3). Interestingly, the lowest infectious viral loads occurred in the animals boosted with NVX-CoV2373, independent of the primary vaccination (Fig. 5j). Sham- or homologous-boosted hamsters showed the highest proportion of breakthrough infections (Fig. 5j).

## Discussion

The emergence of SARS-CoV-2 Omicron subvariants with several amino acid changes in the spike protein has been responsible for extensive reports of breakthrough infections[14–16], mainly related to their increased immune escape by BA.5 and descendants such as BQ.1 and BQ.1.1, which remains one of the most successful Omicron variant at the time of this report. This scenario jeopardizes vaccine efficacy and highlights the need for booster immunizations for all the currently approved vaccines. We evaluated the efficacy of single high and low dose immunization with the Janssen Ad26.COV2.S vaccine and a two series dose of the Pfizer/BioNTech BNT162b2 vaccine, combined with either homologous or heterologous (original mRNA-1273 from Moderna and NVX-CoV2373 from Novavax) booster doses. Immunogenicity and protection against the Omicron BA.5 variant, which was circulating in many parts of the world as the dominant variant between June–November, 2022[17], were evaluated in a hamster model. By including large numbers of animals and realistic timings between primary vaccination, boosters, and infection, this represents one of the most comprehensive preclinical studies of COVID-19 vaccine efficacy yet reported. Our study also utilized all vaccines licensed in the United States, and measured their efficacy in conjunction with novel boosting strategies not yet widely employed. In summary, our study showed greater protection against BA.5 infection when a heterologous booster, especially NVX-CoV2373, was administered in hamsters after primary immunization with either Ad26.COV2.S or the more widely utilized BNT162b2.

While both low- and high-dose primary immunization with Ad26.COV2.S provided significant reductions in BA.5 challenge viral

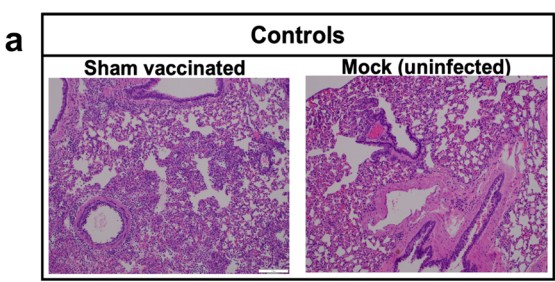

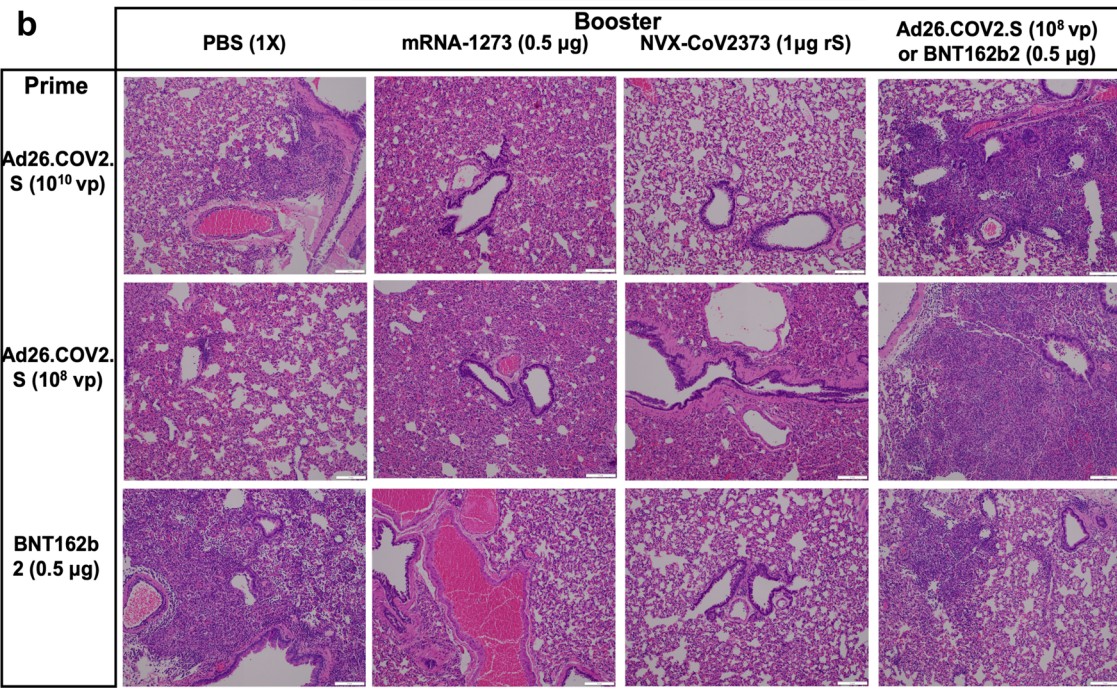

**Fig. 4 | Histopathology of lungs of vaccinated and boosted hamsters challenged with Omicron subvariant BA.5.** Representative hematoxylin and eosin (H&E) images of the left lung of hamsters 4 days after intranasal challenge with $10^4$ PFU of BA.5. **a, b** H&E staining of lung sections harvested from sham-vaccinated (**a**, left; scale bar 100 μM) and mock-infected (**a**, right, scale bar 100 μM) or vaccinated and boosted animals (**b**; scale bar 100 μM) at 4 dpi with BA.5. A section from a sham-vaccinated and a mock-challenged animal are shown for comparison. The representative images are from four lung sections per animal from $n = 4-5$ per group. See Source Data for the complete histopathological evaluation of each specimen.

loads, even after 9 months post-vaccination without boosters, we observed substantial losses of serum neutralizing activity against both the ancestral USA-WA1/2020 and BA.5 SARS-CoV-2 strains, coupled with breakthrough infection and severe pulmonary lesions after BA.5 challenge. As previously shown in humans, neutralizing and binding antibody titers against SARS-CoV-2 induced by two doses of BNT162b2 vaccine peaked at 28–56 dpv and declined by 6 months, and even further by 8 months post-vaccination[18–20]. A single dose of the Ad26.COV2.S vaccine induced substantially lower initial neutralizing antibody than 2 doses of the mRNA vaccine at peak immunity; however, unlike the mRNA vaccine, these Ad26.COV2.S-induced titers remained relatively stable over 8 months[18,21–23]. The stability of the Janssen COVID-19 vaccine's immune response should therefore be considered in future decisions regarding COVID-19 vaccination strategies, especially in populations where cold chains and the difficulty of return visits to health care settings for boosters represent challenges. Furthermore, with the currently updated (bivalent) mRNA vaccines, which in our study showed a better performance to protect against Omicron subvariants when compared with the original versions, an updated version of Ad26.COV2.S should be considered and evaluated. Other studies in NHP[24] and mice[25] showed that a booster with an Omicron BA.1-spike Ad26.COV2.S-based vaccine (Ad26.COV2.S.529) improved the immune responses and showed a marked reduction in lung viral loads and pathology compared with the original vaccine (Ad26.COV2.S). Previous studies using hamsters immunized with Ad26.COV2.S- and BNT162b2 also showed high titers of binding and neutralizing antibodies against the ancestral (USA-WA1/2020) and VOCs (Alpha, Beta and Gamma) at 4 weeks post- vaccination[24–26]. However few studies evaluated the kinetics the antibody response of these vaccines and their efficacy after challenge at 9 months. This longitudinal study therefore represents one of the most comprehensive preclinical investigations analyses of the capabilities of available vaccines, and suggests new strategies to increase their potency.

A double intramuscular 5-μg dose of NVX-CoV2373, a protein-based adjuvanted vaccine, has demonstrated protective human efficacy in the United Kingdom[27], U.S., Mexico[28], and South Africa[29]. On July 13, 2022 the U.S. Food and Drug Administration (FDA) issued an emergency use authorization for this vaccine to be administered as a two-dose primary series, 3 weeks apart and approval was also granted for use as a first booster[30]. Here, we showed that a booster with NVX-CoV2373 vaccine 6 months after primary vaccination with either a single-dose of Ad26.COV2.S or two-doses of BNT162b2 enhanced protection against the Omicron BA.5 subvariant, increasing neutralizing antibody titers to up 3-months-post-booster and significantly decreasing viral loads in challenged infected hamsters. However, many animals still developed detectable challenge infections (Fig. 5), highlighting the difficulties in completely eliminating infection and transmission through vaccination. Recently, findings in clinical studies also

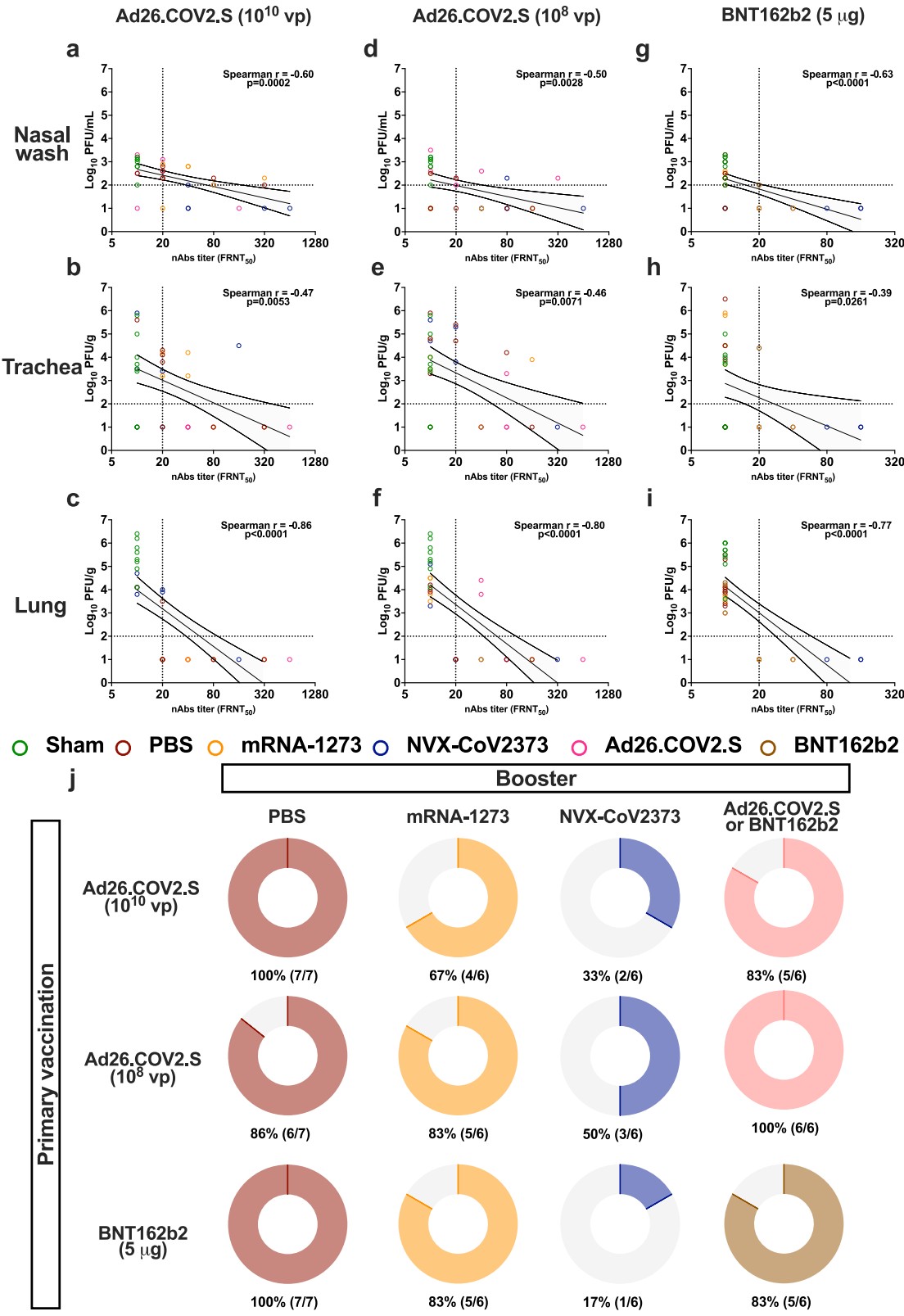

showed that after a third dose of NVX-CoV2373, high neutralization titers against various Omicron subvariants, including BA.1 and BA.4/BA.5, were observed[31]. Based on our results and others, the NVX-CoV2373 vaccine should be strongly considered as a booster worldwide.

Our study has several limitations. First, it has been demonstrated that the Omicron variants[32,33], including BA.4/BA.5[34], are less

pathogenic in hamsters than other VOCs or the ancestral SARS-CoV-2 strain. This could lead to an inability to compare protection offered against BA.5 compared to more virulent strains in this animal model. Second, our experiments were performed exclusively in male hamsters to ensure sufficient power in the various experimental cohorts. Further studies in this and other animal models should include both males and females. Third, our measures of immunogenicity focused on the

**Fig. 5 | Correlation between neutralizing antibody titers and protection against BA.5. a–i** Correlation between the FRNT$_{50}$ neutralization titers at 3 months post-boost (pre-challenge) against BA.5 (x-axis) and the infectious virus loads (y-axis) detected at 4 dpi in (**a, d, g**) nasal washes, (**b, e, h**) trachea, and (**c, f, i**) lungs of hamsters primed with (**a–c**) one $10^{10}$ vp dose of Ad26.COV2.S (n = 6 or 7 per group), (**d–f**) one $10^8$ vp dose of Ad26.COV2.S (n = 6 or 7 per group), or **g–i** two 5 μg doses of BNT162b2 (n = 6 or 7 per group). **j** Proportion of hamsters developing break-through infections following BA.5 challenge. Percentage (%) and number of animals with detectable infectious virus loads in at least one sample (nasal wash, trachea, and/or lungs) are shown below each graph. Colors designate experimental cohorts (n = 6–7 animals per group). In panels **a–i**, data points represent individual animals, thick lines represent linear regression, dashed lines with shading represent 95% CI, and dotted lines represent the lower limit of detection for each assay. Spearman's rank correlation coefficient (r) and p-values are shown. In panel **j**, the color-coded portion of the ring represents the proportion of hamsters developing a break-through infection following BA.5 challenge. Panels were generated using GraphPad Prism v9.4 software. See Source Data for complete data.

neutralizing antibody responses and did not account for possible cross-reactive binding antibodies and T cell responses, which could impact protective immunity. Fourth, due to the large number of animals in the study we were only able to analyze a single concentration of the NVX-CoV2373 booster, which could lead to an overestimation of protection. In addition, the low dose of mRNA-1273 vaccine used was designed to model individuals with suboptimal immune responses, which may impact its comparison with other optimal vaccines doses. The doses of all vaccines, determined based on immune responses similar to those seen in humans and experience within the NIH-SAVE network[35] on rodent vaccinations, also may not be truly comparable. In reality, when expressed as a fractional human dose, all rodent studies have used very high doses. We selected our doses to be similar what has been used in previous hamster studies so that our data are comparable. Further studies using lower doses and bivalent original/Omicron BA.4-5 mRNA vaccines might be needed to assess the enhanced neutralizing antibody responses and protection against BA.5. Finally, due to the number of animals per subgroup and the number of hamsters harvested each time-point (2 and 4 dpi), it was only possible to challenge the animals with one Omicron subvariant. At the time of our study, BA.5. represented the majority of new cases and posed the single greatest threat and was therefore the focus of our efforts. Currently BQ.1 and BQ1.1, descendants of BA.5, are increasing in incidence[9]. Future studies should replicate our work in these newer strains.

In conclusion, we showed the kinetics of the humoral response to the Ad26.COV2.S and BNT162b2 COVID-19 vaccines in Syrian hamsters for up to 9 months, and we evaluated the effect of several boosters. The Ad26.COV2.S at either low or high doses provided relatively stable neutralizing antibody levels, and the heterologous boost with NVX-CoV2373 vaccine resulted in a substantial increase in humoral immune responses and the most robust protection against Omicron BA.5 infection. These data contribute to our understanding of heterologous and homologous COVID-19 vaccine strategies and can contribute to recommendations for vaccination of humans.

## Methods

### Ethics statement

All animal experiments and procedures were performed in accordance with the guidance for the Care and Use of Laboratory Animals of the University of Texas Medical Branch (UTMB). The protocol was approved by the Institutional Animal Care and Use Committee (IACUC) at UTMB (protocol number 2009087). Injections, bleeds and nasal washes were performed under anesthesia that was induced and maintained with isoflurane (Covetrus, Portland, ME, USA). All efforts were made to minimize animal stress and suffering for the duration of the study. The Institutional Biosafety Committee (IBC) approved work with infectious SARS-CoV-2 under BSL3 conditions.

### Cells and viruses

The SARS-CoV-2 USA_WA1/2020 (SARS-CoV-2/human/USA/WA-CDC-02982586-001/2020, GenBank: MN985325.1) and B.1.1.529 (Omicron) sublineage BA.5.5 (hCoV-19/USA/COR-22-063113/2022, GISAID accession ID: EPI_ISL_13512579) were obtained from the World Reference Center for Emerging Viruses and Arboviruses (WRCEVA, University of Texas Medical Branch, Galveston, TX, USA). Both viruses were passaged once in VeroE6-TMPRSS2 cells for subsequent experiments and deep sequenced to verify single nucleotide variants (SNVs) frequencies along the whole genome at the UTMB's Next Generation Sequencing (NGS) Core, directed by Dr. Steven G. Widen. Deep sequencing of these stocks revealed no mutations or deletions in the Spike protein >5.0% frequency.

VeroE6 cell line expressing human TMPRSS2 were obtained from JCRB Cell Bank (JCRB1819, lot 04172020)[36]. VeroE6-TMPRSS2 cells were maintained in Dulbecco's modified Eagle's medium (DMEM; Gibco/Thermo Fisher Scientific, Waltham, MA, USA) supplemented with 10% fetal bovine serum and 100 μg/ml of Geneticin™ Selective Antibiotic - G418 Sulfate (Thermo Fisher Scientific). VeroE6-TMPRSS2 cell line was verified and tested negative for mycoplasma before use.

### Vaccines and viral antigens

As primary dose we used the Ad26.COV2.S (Janssen-Johnson & Johnson) vaccine that is a recombinant, replication-incompetent human adenovirus type 26 vector[37] encoding full-length SARS-CoV-2 spike protein in a prefusion-stabilized conformation[38] and the BNT162b2 (Pfizer-BioNTech) vaccine[39] that is a lipid nanoparticle–formulated[40], nucleoside-modified RNA vaccine[41] that encodes a prefusion stabilized, membrane-anchored SARS-CoV-2 full-length spike protein[42]. Ad26.COV2.S and BNT162b2 vaccines were acquired from UTMB Health Services vaccine clinics after the human doses had been used and keep frozen at −80 °C until hamster's vaccination. Remnant vaccine from each vial was pooled and injected into animals within 1 h after thaw.

Regarding the heterologous vaccines used to boost the animals, we used the preclinical mRNA-1273 vaccine (Moderna), a lipid nanoparticle–encapsulated mRNA-based vaccine[43–45] that encodes the prefusion stabilized full-length spike protein of the Wuhan-Hu-1 (ancestral) SARS-CoV-2 and the NVX-CoV2373 vaccine (Novavax), or a recombinant nanoparticle vaccine[46] against SARS-CoV-2 that contains the full-length spike glycoprotein of the prototype strain plus Matrix-M adjuvant[47]. The mRNA-1273 and NVX-CoV2373 vaccines were provided directly by the companies and stored until further use at −20 °C and 4 °C, respectively. The Ad26.COV2.S[26,48], BNT162b2[49], mRNA-1273[50] and NVX-CoV2373[51] vaccines doses used in the study were chosen based on previous studies that showed the similarities in the antibody levels induced by vaccination of mice or hamsters to those generated in vaccinated humans.

SARS-CoV-2 WT recombinant spike S1 protein, used to perform the ELISAs, was purchased from Sino Biological (Cat. # 40591-V08H). S1 protein were produced in hexahistidine-tagged 293HEK cells and contains the same sequence as USA_WA1/20 strain.

### Animal experiments

Two groups of 50, 4–5-week-old male Syrian hamsters (*Mesocricetus auratus*, HsdHan: AURA strain, Envigo, Indianapolis, IN), were vaccinated with $1 \times 10^8$ or $1 \times 10^{10}$ viral particles of Ad26.COV2.S vaccines delivered intramuscularly in one 100 μL dose in the hind leg. Another group of 50 were immunized 3 weeks apart with 5 μg of BNT162b2

vaccine (100 μL/dose). Additionally, a placebo (PBS 1X) group of 59 animals were included (Supplementary Fig. 2). At the time of primary vaccination, 4–5 week-old male Syrian hamsters were healthy and with a normal average weight between 40–60 g. Blood sample was collected via the retro-orbital plexus under isoflurane anesthesia every 28 days post-vaccination (dpv) up to 168 dpv ($n = 20$ per group) for immunogenicity analysis. Samples were centrifuged at 2000 g for 10 min for serum separation, aliquoted and stored at −20 °C. Six months (168 days) after completing the primary series immunization, all animals ($n = 208$) were bled for neutralizing antibody quantification, and then boosted with different vaccines. All of the three groups vaccinated with $10^{10}$ vp Ad26.COV2.S, $10^8$ vp Ad26.COV2.S or 5 μg of BNT162b2 vaccines were heterologous boosted with the same vaccines: mRNA-1273 (0.5 μg) and NVX-CoV2373 (1 μg rS/15 μg Matrix-$M^{TM}$). A homologous booster was also used, $10^8$ vp Ad26.COV2.S for both groups primary vaccinated with Ad26.COV2.S vaccine and 0.5 μg of BNT162b2 for the group primed with this vaccine. In additional, a PBS boosted group was included for comparison. Supplementary Fig. 2 shows the vaccination regimens (prime + booster) used in the study.

Four weeks and 3 months (pre-challenge) after boost, all animals were bled for antibody analysis. The hamsters were challenged with $10^4$ PFU/dose of Omicron BA.5 diluted in sterile DPBS (100 μL/dose), delivered intranasally and equally split between each nostril. We used $10^4$ PFU as the challenge dose, based on our previously experience with others Omicron subvariants and based on other studies from the NIH-SAVE program[33,52,53], who showed that a $10^4$ PFU is a dose that enables controlled investigation of pathogenesis, correlates of protection and efficacy of vaccination in the hamster model.

Weights and clinical score were recorded daily until 4 days post infection (dpi). Scoring (from 1–4) was based on the following criteria, 1 for healthy animals (bright, alert and reactive), 2 for ruffled fur and/or lethargic, 3 for a score of 2 plus 1 additional clinical signs such as, hunched posture, orbital tightening, or neurological signs such as ataxia/tremors/paralysis of 1 limb, and/or >10% weight loss and 4 for a score of 3 plus 1 additional clinical sign such as, reluctance to move when stimulated, or neurologic signs (seizures, paralysis of >1 limb, inability to right itself) or >20% weight loss. On day 2 and 4 post-infection, nasal washes were performed and between 5–8 animals from each group were euthanized per day. The lungs and trachea were excised, weighed, and samples were taken for virological, immunological, and histopathological analyses.

## Plaque assay

Excised trachea sections and right cranial lung lobe tissue were weighed and homogenized using a TissueLyser II (Qiagen). Briefly, fresh trachea and lung tissue were placed in a 2.0 ml microtube containing 0.5 ml cold DMEM supplemented with 2% FBS and 1% of geneticin (1 mg/ml), and one 5 mm stainless steel bead. Sample tubes were homogenized at 26 Hz for 1 min, then centrifuged to pellet debris (4 °C, 5 min at ~21,000 x g) and the supernatant was collected and stored at −80 °C to be used for viral plaque assay.

For virus titration, approximately $5 × 10^5$ VeroE6-TMPRSS2 cells were seeded in 12-well plates and cultured at 37 °C, 5% $CO_2$ for 24 h. Virus was 10-fold serially diluted in DPBS and 100 μl of diluted viruses were transferred onto the monolayers. The viruses were incubated with the cells at 37 °C with 5% $CO_2$ for 1 h. After the incubation time, overlay medium was added to the infected cells. The overlay medium contained DMEM (2X) with 5% FBS, 1% geneticin and 0.8% sea-plaque agarose (Lonza, Walkersville, MD). After 48 h of incubation, plates were stained with 0.03% of neutral red (Sigma-Aldrich, St. Louis, MO) for 6 h and then plaques were counted using a lights box. The viral titers from nasal washes were expressed as PFU/ml, calculated as [(number of plaques per well) × (dilution factor)]/(inoculum volume) and from trachea and lung tissues as PFU/g, calculated as {[(number of plaques per well) × (dilution factor)]/(inoculum volume)} × tissue

weight. The limit of the detection (LOD) of the assay is $1 × 10^2$ PFU/ml or PFU/g.

## Plaque reduction neutralization test (PRNT)

Approximately $5 × 10^5$ VeroE6-TMPRSS2 cells were seeded to each well on 12-well plates and cultured at 37 °C, 5% $CO_2$ for 24 h. Sera were heat inactivated at 56 °C for 30 min prior to being diluted 1:10 in PBS 1X. Serial two-fold dilutions (1:10 to 1:1280) were performed in PBS with 2% FBS. An equal volume of virus diluted to 800 PFU/ml was added to each dilution (final dilution as 1:20 to 1:2560). Following a 1-h incubation at 37 °C, 5% $CO_2$, cell monolayers were infected with 100ul of each dilution for another hour at 37 °C, 5% $CO_2$. After the incubation time, overlay medium (DMEM (2X) with 2% FBS, 1% geneticin and 0.8% sea-plaque agarose) was added to the infected cells. After 48 h incubation, plates were stained with 0.03% neutral red (Sigma-Aldrich) and plaques were counted on a light box. The neutralization titers were calculated and expressed as the reciprocal serum dilution yielding ≥50% reduction ($PRNT_{50}$) in the number of plaques as compared to control wells. The LOD of the test was set as 1:20.

## Focus reduction neutralization test (FRNT)

FRNT was adapted from previously described protocols[54,55]. Briefly, $3 × 10^5$ VeroE6-TMPRSS2 cells/mL were seeded in 96-well plates and cultured at 37 °C, 5% $CO_2$ for 24 h. Sera were heat inactivated at 56 °C for 30 min prior to be diluted. Serial two-fold dilutions (1:10 to 1:1280) were performed in DPBS. An equal volume of virus diluted to $5 × 10^3$ FFU/ml was added to each dilution (final dilution as 1:20 to 1:2560). Following a 1-h incubation at 37 °C, 5% $CO_2$, cell monolayers were infected with 20 μl of each dilution for another hour at 37 °C, 5% $CO_2$. After the incubation time, overlay medium of 85% MEM (Gibco, Grand Island, NY) and 15% DMEM supplemented with 1% antibiotic-antimycotic and 0.85% methyl cellulose (Sigma, St Louis, MO) was added to the infected cells. After 36 h incubation, the monolayers were fixed with formalin (Fisher, Pittsburgh, PA) for at least 24 h. Following fixation, plates were washed twice with DPBS (Sigma, St Louis, MO) and 100 μL of permeabilization buffer (PB) consisting of DPBS supplemented with 0.1% BSA (Sigma, St Louis, MO) and 0.1% saponin (Sigma, St Louis, MO) was added for 30 min at room temperature. PB was removed, and monolayers were incubated overnight at 4 °C with rabbit polyclonal antibody against SARS-CoV N protein (Dr. Shinji Makino, Department of Microbiology & Immunology, UTMB, Galveston, TX) diluted in PB at a ratio 1:3000. Excess antibody was removed by washing with DPBS, and monolayers were incubated for 1 h at room temperature with HRP-conjugated goat anti-rabbit IgG (Cat. # 7074, Cell Signaling, Beverly, MA) diluted as 1:2000 in PB. Excess antibody was washed away with DPBS, and foci were stained using KPL TrueBlue Peroxidase Substrate (Cat. # 5510-0050, SeraCare, Milford, MA). Once foci were visible under a light microscope excess substrate was removed and the monolayers were washed with water. Wells were visualized and imaged using the Cytation7 Imagining Reader (BioTek, Winooski, VT). Foci were counted manually and the neutralization titers were calculated and expressed as the reciprocal serum dilution yielding ≥50% reduction ($FRNT_{50}$) in the number of foci as compared to control wells. The LOD of the test was set as 1:20.

## Enzyme-linked immunosorbent assay (ELISA)

ELISAs to evaluate the IgG binding to SARS-CoV-2 spike subunit 1 (S1) were adapted from a previously described and validate assay[56]. Briefly, 96-well plates (Immulon 4 HBX; Thermo Fisher Scientific) were coated overnight at 4 °C with recombinant S1 proteins at 2 μg/ml in PBS 1X (Gibco). Plates were washed 3 times with 300 uL of PBS + 0.1% Tween 20 (PBS-T) and 200 μl per well of 3% non-fat milk prepared in PBS-T was added to the plates and incubated at 37 °C for 2 h. Serum samples were heated at 56 °C for 30 min before use. Serum diluted 1:100 was further 3-fold serially diluted in PBS-T and incubated at 37 °C for 1 h. Plates were

washed three times with PBS-T and then incubated with goat anti-hamster IgG (Cat. # PA1-29626, Thermo Fisher Scientific) secondary antibody conjugated to horseradish peroxidase (HRP) in PBS-T with 1% non-fat milk at 1:7000 dilution, respectively. After 1 h of incubation at 37 °C, plates were washed 3 times with 300uL of PBS-T and 100 μl of SIGMAFAST OPD (o-phenylenediamine dihydrochloride; Sigma–Aldrich) solution was added to each well. The reaction was stopped by the addition of 100 μl per well of 3 M HCl and the optical density at 490 nm ($OD_{490}$) was measured in a microplate reader (VersaMax, Molecular Devices). The threshold for positivity was set at $OD_{490}$ values higher than the negative control plus 3 standard deviations in at least two consecutive dilutions. A positive control (obtained from infected hamsters in the convalescent phase) and a negative control (obtained from unvaccinated hamsters prior to the start of the study) were added to every assay plate for validation. The average of its signal was used for normalization of all the other values on the same plate.

## Histology

Left lungs at day 4 post-infection were harvested from hamsters and fixed in 10% neutral buffered formalin (NBF) for at least 7 days. After buffer exchange, fixed tissue was embedded in paraffin, cut into 5 μM sections, and stained with hematoxylin and eosin (H&E) on a SAKURA VIP6 processor by the Anatomic Pathology Laboratory at University of Texas Medical Branch. The lung sections were blinded analyzed and full evaluated by a pathologist. Visual observations of lung abnormalities were scored from 0–4, reflecting the severity of abnormalities in infected hamsters compared to control hamsters. Each section was analyzed for degree of involvement and scored as 0 (none, 0%), 1 (minimal, 1–25%), 2 (mild, 26–50%), 3 (moderate, 51–75%), or 4 (severe, 76–100%)[57]. The photomicrographs were captured with an Olympus BX43 microscope mounted with an Olympus DP27 digital camera. All images were analyzed using Olympus cellSens Entry 3.2 software and taken at 10X magnification.

## Statistical analysis

All data were analyzed with GraphPad Prism version 9.3 software. Tests, number of animals, median values, and statistical comparison groups are indicated in the Fig. legends. Changes in infectious virus titer, or serum antibody responses were compared to sham-vaccinated animals and were analyzed by one-way ANOVA with a multiple comparisons correction, Mann–Whitney test, or Wilcoxon signed-rank test depending on the type of results, number of comparisons, and distribution of the data. A p-value of <0.05 indicates statistically significant.

## Reporting summary

Further information on research design is available in the Nature Portfolio Reporting Summary linked to this article.

## Data availability

All data supporting the findings of this study are available within the article and its Supplementary Information and Source Data. Source data are provided with this paper.

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

## Acknowledgements

This project has been funded in whole or in part with Federal funds from the National Institute of Allergy and Infectious Diseases, National Institutes of Health, Department of Health and Human Services, under Contract No. 75N93021C00016 and grant R24 AI120942 (to S.C.W.), and by the Sealy and Smith Foundation (to SCW). During part of the study R.R.G.M. received a grant from Fundação de Amparo à Pesquisa do Estado de São Paulo—FAPESP (2019/27803-2), W.M.S. is supported by a Global Virus Network fellowship. We thank the in vivo NIH-SAVE group for their thoughtful insights during the study, Nehad Saada, Allan McConnell and Patricia A. Valdes for technical assistance. We acknowledge the Anatomic Pathology Laboratory at University of Texas Medical Branch for tissue sectioning. The authors acknowledge Mimi Guebre-Xabier, Nita Patel, Gale Smith, and Greg Glenn at Novavax, Inc. for providing NVX-CoV2373 vaccine and Sayda M. Elbashir and Darin Edwards at Moderna, Inc. for providing the preclinical version of mRNA-1273 vaccine.

## Author contributions

Conceptualization, R.R.G.M., K.S.P. and S.C.W; Methodology, R.R.G.M. and J.A.P.; Formal analysis, R.R.G.M.; Investigation, R.R.G.M., J.L.W., D.S., G.H.R., B.M.M., R.A.R., W.M.d.S., J.L., J.A.P., D.H.W. and K.S.P.; Resources, K.S.P. and S.C.W.; Data curation, R.R.G.M.; Writing—Original draft, R.R.G.M.; Writing—Review and Editing, J.A.P., K.S.P., and S.C.W.; Visualization, R.R.G.M.; Supervision, K.S.P. and S.C.W.; Funding acquisition, S.C.W.

## Competing interests

The authors declare no competing interests.
