## [Peer Review File · Nature Communications]

Immunogenicity and efficacy of vaccine boosters against SARS-CoV-2 Omicron subvariant BA.5 in male Syrian hamstersREVIEWER COMMENTS

Reviewer #1 (Remarks to the Author):

Machado and colleagues present a unique long-term preclinical vaccination study using the hamster model comparing three main human Covid-19 vaccines in wide clinical use and heterologous booster combinations thereof. The main strength of the study are (i) the broad scope (multiple homologous and heterologous combinations) and (ii) the consistent use of immunogenicity and efficacy testing against the relevant BA.5 virus, thus challenging first generation vaccines that are based on "historical" 2019 Wuhan spike by viruses that dominate current 2022/2023 clinical reality. The study is hence timely and high relevant. The presented data are comprehensive and sound. The discussion considers limitations of the study, but can be enhanced and include some extra issues. Some figure panels are difficult to read and may benefit from a change in the color pallet.

Specific comments:

- Abstract, line 38 and later throughout the text: „complete protection“ needs to be defined (protection from what exactly with many options: infection, cytokine elevation, antibody recall, lung pathology, weight loss, etc.). As reported in lines 320-330, there was no "complete" protection in any tested condition, if at all marked reduction in viral loads. Lung lesions remained.
- Line 92 and throughout text: We are not fully sure but to the best of our knowledge the correct species name is "Syrian golden hamster", or simply "Syrian hamster" or "Golden hamster". I suggest simplifying throughout text yet to include the scientific name *Mesocricetus auratus* as well; here or in the M&M section.
- Line 98 & 100: Primary vaccination was done in weaned/young hamsters (4-5 wks old). Neither here nor in the M&M their weight is mentioned, also consider how the used doses relate to the clinically used human doses (absolutely and/or per bodyweight), or likewise to the human pediatric doses. How do the doses used here relate to the doses used in other preclinical studies in the hamster model.
- Line 117-131: I suggest to include next to reporting GMTs, also 95% CI and a mention of the fraction of seropositive animals. To us it appears as if the spread of data and in particular the number of animal losing any detectable antibody over time is markedly different between groups. Same suggestion applies also to booster study. Considering the vast amount and rich set of data it maybe advisable to include a summary table reporting key GMT plus 95%CI, IgG AUC and fraction of animals seroconverted for the different groups at selected timepoints for more simple comparison of different regimens post primary, pre-boost and post-boost (i.e. on top of the full record of all experimental data provided in the Extended data/Supplementary Tables).
- Fig-1: Please mention group sizes in legend. Are the AUC data presented in E-G derived from those shown in H-J? In that case it would be more appropriate to change the order of presentation (first original, then derived data)
- In panels H-J lines are very thin and colors coding for individual timepoint difficult to distinguish. Please consider e.g. to use another color pallet. Also the color used in H-J are same as used before to distinguish individual vaccines, rather than time points.
- Line 165: Please mention whether the sham group was age-matched.
- Line 175-191: In contrast to paragraph before, while discussing fold-changes in GMTs no p-values are reported anymore. Does this mean no significance any more?
- Line 199: twice "wherein"
- Figure 2: Color pallet really difficult to read/conditions difficult to distinguish, particularly in E-F.
- Line 234: Infection with earlier isolates of SARS-CoV2 had a poor dose-response (almost no matter how high or low; always saturating regarding replication in tissues). This has changed for Omicron. How does the dose used (10e4 PFU) related to what has been used and benchmarked by others before.
- Line 268: Considering that virus loads are reduced in trachea and lungs by orders of magnitude, it maybe more appropriate to round numbers (e.g. 690,000 or 700,000 instead of 686,889) calculated for fold changes.
- Line 264-289: Alike for summary table GMT etc., I also suggest to include a summary table

reporting fold-changes for viral loads in the different groups for the key message; namely whether or not infection outcomes differ between vaccine groups. Only few people will dig into the Supplementary tables to extract these data themselves.

- Figure 3C-E: Sham groups for each first two panels (left and middle) seem to be identical and different from third (right). Please explain. It is somehow trivial that virus load in all vaccinated animals differ significantly from sham (as indicated). A key question is however, how do infection outcomes differ between vaccine groups? Can this be implemented in the figures (or new table as discussed above).
- Figure 4: I suggest to include a graphical presentation of histological scores (bar graph, spider plot, etc.) in the different groups for ease of comparison. Please include arrows or other symbols to highlight relevant pathologic changes (as mentioned in the results narrative: moderate to severe interstitial pneumonia characterized by hemorrhages, perivascularitis, perivascular edema, arterial endothelial leukocytic margination, peribronchiolitis, vasculitis and bronchiolitis).
- Line 390: Statement many to be updated e.g. "BA5 and descendants such as BQ.1Q"
- Line 408: "significant protection" to be defined. Neither of the used Covid-19 vaccines in any combination protected with high efficacy against infection and pathology in hamsters.
- Line 417-420: I agree that based on the immunogenicity and efficacy data presented herein, Ad26 is not inferior to other platforms such as mRNA. However, this statement regarding the Janssen vaccine may be a bit too strong. (i) The number of breakthrough cases in the 2x high Ad26 group is equally good (bad) as in 2x mRNA and worse than high Ad28 + mRNA (neglecting lack of statistical power due to the small groups). (ii) Also Janssen uses a historical, mismatched antigen, whereas already updated (bivalent) mRNA vaccines are available. May the discussion hence not rather endorse an update to the promising yet old Ad26 vaccine?
- Line 437: "increasing neutralizing antibody titers to up 3-months-post-booster and significantly decreasing viral loads in challenged infected hamsters". This inverse correlation between nAb and viral loads is suggested in Figure 5. How strong do the authors judge the predictive power of nAb levels regarding protection against BA.5. From the data it appears more like "when nAb detectable than >90% chance to have lower viral load" thus more digital response. Are there clinical implications that could be drawn, also considering that data for 2 dpi (Figure S5) look much less promising as many more breakthrough infections can be observed?
- Line 446: ... overestimation of <the level of> protection ...
- Line 448: What is the possible skew in outcome the authors expect from using solely males?
- Regarding immune reaction, or pathology? Please give appropriate references.
- Line 453-459: The discussion of the doses used per vaccine should be more explicitly comparing doses used in previous studies in hamsters and in humans. E.g. what to conclude from the application of the 1/10 of a human dose in hamsters that weigh 1000x less than a humans. What is the evidence that these preclinical data translate into the clinics.
- Line 671: How were the lung sections scored and by whom (blinded/unblinded)?
- Line 950: Please mention in legend that presented Ab titers relate to prototypic strain.
- Line 967: What is the relevance/meaning of the here mentioned 90% cutoff, like a euthanasia
- Data given in extended data tables suffer from some lack of information in the legends (what are all the different values given in brackets?) and inconsistent use of "," and "." for numeric values.

Reviewer #2 (Remarks to the Author):

Summary:

Machado et al. compare and contrast an experimental vaccine study in hamsters in which a series of homologous and heterologous prime (Pfizer/Janssen) / boost (Moderna, Novavax) vaccine regimens is compared, followed by challenge with BA.5. They track neutralizing antibody responses in the plasma of each cohort, as well as examining in vivo protection against weight loss, viral load, and lung histopathology.

Points of concern:

- The section spanning line 175-191, describing the boost neut data in Figure 2, is incredibly confusing and difficult to coordinate with the main figure.
 - o The NVX boost is discussed first but is found in the third column.
 - o In general, the paired analyses found in Fig. 2A-2C should likely be arranged in a group analyses similar to 2D-2F. It is difficult to track with the statements made in the main text.
 - o Line 190: Should BNT162b2 actually be AD28-10⁸?
 - o Line 187: The text goes between saying 3 months post-boost and day 252. This is confusing.
 - o Lines 193-202 I cannot understand. Without a figure to guide the analysis, attempting to interpret is very difficult.
- Once again in Figure 3, the figures are incredibly difficult to analyze due to size/resolution and poor color contrast for data series; and the supplementary tables are formatted poorly to enable a reviewer to reasonably check. Figures HAVE to be more interpretable.
 - o It appears the NVX boost is the winner in regards to challenge experiments. Lines 264-266 have this stated, but only refer to Fig. 3C. This should be Fig. 3C-3E.
 - o It appears that the PBS boosts series for Ad26 prime data are quite similar to homologous/heterologous boosts in regards to weight loss. I don't think this was addressed in the main text and seems important in light of interpreting differences among the homologous/heterologous boosts.
 - o Lines 274-277 address the above with the viral titer data. In general, this is a major point in regards to interpreting the boost data collectively. The figure (and main text) unfortunately seem to bury such findings in a mountain of other data.
 - o Overall, this section of the main text poorly coordinates with the main figure. There is a lot of data here. Coordinating the results main text with an interpretable figure is a must and the bar is not meant here in that regard.
 - o Figure 3A challenge virus label appears to be incorrect
- Section comprising lines 315-331, describing histology in Figure 4 seems to indicate conflicting descriptions of the histology. Lines 322-323 states "No apparent differences were observed among groups receiving different vaccines or boosters", yet some differences are described throughout the section. However, I am not visually able to discern the described differences in the Figure 4 images, albeit I do not consider myself well trained in detected small differences in histopathology of lung tissue. With that said, if differences are meant to be readily discerned from these images, labels or a more concrete scoring method would be useful.

Response to Reviewer #1 Comments

Reviewer #1 (Remarks to the Author):

Machado and colleagues present a unique long-term preclinical vaccination study using the hamster model comparing three main human Covid-19 vaccines in wide clinical use and heterologous booster combinations thereof. The main strength of the study are (i) the broad scope (multiple homologous and heterologous combinations) and (ii) the consistent use of immunogenicity and efficacy testing against the relevant BA.5 virus, thus challenging first generation vaccines that are based on “historical” 2019 Wuhan spike by viruses that dominate current 2022/2023 clinical reality. The study is hence timely and high relevant. The presented data are comprehensive and sound. The discussion considers limitations of the study, but can be enhanced and include some extra issues. Some figure panels are difficult to read and may benefit from a change in the color pallet.

We thank the reviewer for the positive feedback regarding our study. We have reviewed the entire manuscript, adhering to all the proposed suggestions to enhance the quality of the work. All changes in the manuscript are color highlighted in yellow. In addition, we have changed the color pallet and increased the figure resolution to 600 dpi for a better visualization of the data. Color schemes were updated following *Nature Communications* formatting instructions (<https://www.nature.com/documents/ncomms-formatting-instructions.pdf>), avoiding rainbow gradients (red and green simultaneously) and choosing color-blind-friendly pallets.

Specific comments:

- Abstract, line 38 and later throughout the text: „complete protection“ needs to be defined (protection from what exactly with many options: infection, cytokine elevation, antibody recall, lung pathology, weight loss, etc.). As reported in lines 320-330, there was no “complete” protection in any tested condition, if at all marked reduction in viral loads. Lung lesions remained. We thank the reviewer for spotting this issue. We have replaced “complete protection” with “reduction in BA.5 viral replication,” in the abstract (line 37).

- Line 92 and throughout text: We are not fully sure but to the best of our knowledge the correct species name is “Syrian golden hamster”, or simply “Syrian hamster or “Golden hamster”. I suggest simplifying throughout text yet to include the scientific name *Mesocricetus auratus* as well; here or in the M&M section.

We thank the reviewer for catching this error and we have uniformly corrected the species name to “Syrian hamster” following NCBI Taxonomy (<https://www.ncbi.nlm.nih.gov/Taxonomy/Browser/wwwtax.cgi?id=10036>). The scientific name *Mesocricetus auratus* is stated in the M&M section (lines 517-518) per the reviewer’s concern.

- Line 98 & 100: Primary vaccination was done in weaned/young hamsters (4-5 wks old). Neither here nor in the M&M their weight is mentioned, also consider how the used doses relate to the clinically used human doses (absolutely and/or per bodyweight), or likewise to the human

pediatric doses. How do the doses used here relate to the doses used in other preclinical studies in the hamster model.

Thank you for your comment, we addressed this by adding a sentence in the M&M section regarding the hamster ages (4-5 wks old) and initial weights (lines 522-523). The doses were not adjusted for body weight and were determined based on previous preclinical studies in mouse and hamster models, as stated in lines 506-509: "The Ad26.COVID2.S23^{26,27}, BNT162b2²⁸, mRNA-1273⁵⁰ and NVX-CoV2373⁵¹ vaccines doses used in the study were chosen based on previous studies that showed similarities in the antibody levels induced by vaccination of mice or hamsters to those generated in vaccinated humans".

- Line 117-131: I suggest to include next to reporting GMTs, also 95% CI and a mention of the fraction of seropositive animals. To us it appears as if the spread of data and in particular the number of animals losing any detectable antibody over time is markedly different between groups. Same suggestion applies also to booster study. Considering the vast amount and rich set of data it maybe advisable to include a summary table reporting key GMT plus 95%CI, IgG AUC and fraction of animals seroconverted for the different groups at selected timepoints for more simple comparison of different regimens post primary, pre-boost and post-boost (i.e. on top of the full record of all experimental data provided in the Extended data/Supplementary Tables).

We thank the reviewer for the suggestion, and as such we added a summary table reporting key GMT plus 95% CI, IgG AUC and fraction of animals seroconverted for the different groups at selected timepoints (post primary, pre-boost and post-boost) as Supplementary Data 1 and 3.

- Fig-1: Please mention group sizes in legend. Are the AUC data presented in E-G derived from those shown in H-J? In that case it would be more appropriate to change the order of presentation (first original, then derived data)

We added the group sizes in the Fig. 1 legend as requested, and we also reference supplementary Fig. 2 where the reader can find more detail regarding group sizes (lines 110-112). Since AUC data presented in E-G are derived from those shown in H-J, we changed the order of presentation as suggested (Fig. 1e-j) for clarity.

- In panels H-J lines are very thin and colors coding for individual timepoint difficult to distinguish. Please consider e.g. to use another color pallet. Also the color used in H-J are same as used before to distinguish individual vaccines, rather than time points.

We increased the thickness of the lines as much as possible to prevent overlap (from 2pt to 4pt), and changed the color pallet used in the H-J panels for better visualization of the data. As suggested, we did not use any of the colors (bordeaux, dark blue and orange) used before in B-G panels.

- Line 165: Please mention whether the sham group was age-matched.

We added this information as requested (lines 137-138): "An age-matched, sham-boosted (PBS) group was included for comparison".

- Line 175-191: In contrast to paragraph before, while discussing fold-changes in GMTs no p-values are reported anymore. Does this mean no significance any more?

We added the requested p-values that were missing (lines 154-165).

- Line 199: twice “wherein”

We thank the reviewer for spotting this typo and have corrected as requested (line 173).

- Figure 2: Color pallet really difficult to read/conditions difficult to distinguish, particularly in E-F.

We have changed color pallet and included a matched colorful bar border to facilitate the distinction of data. Color schemes were updated utilizing *Nature Communications* formatting instructions (<https://www.nature.com/documents/ncomms-formatting-instructions.pdf>), avoiding rainbow gradients (red and green simultaneously) and choosing color-blind friendly pallets.

- Line 234: Infection with earlier isolates of SARS-CoV2 had a poor dose-response (almost no matter how high or low; always saturating regarding replication in tissues). This has changed for Omicron. How does the dose used (10e4 PFU) related to what has been used and benchmarked by others before.

We decided to used 10e4 PFU as the challenge dose, based on our previously experience with others Omicron subvariants and based on other studies from the NIH-SAVE program, such as Halfmann et al., 2022 (<https://doi.org/10.1038/s41586-022-04441-6>), McMahan et al., 2022 (<https://doi.org/10.1016/j.medj.2022.03.004>) and Zhang et al., 2022 (<https://doi.org/10.1038/s41392-022-00930-2>), who showed that a 10e4 PFU is a dose that enables controlled investigation of pathogenesis, correlates of protection and efficacy of vaccination in the hamster model. We have included this information in the M&M section (lines 541-544).

- Line 268: Considering that virus loads are reduced in trachea and lungs by orders of magnitude, it maybe more appropriate to round numbers (e.g. 690,000 or 700,000 instead of 686,889) calculated for fold changes.

We thank the reviewer for the suggestion, we have changed the -fold change values to round numbers (line 240).

- Line 264-289: Alike for summary table GMT etc., I also suggest to include a summary table reporting fold-changes for viral loads in the different groups for the key message; namely whether or not infection outcomes differ between vaccine groups. Only few people will dig into the Supplementary tables to extract these data themselves.

We thank the reviewer for the suggestion, and we included the requested summary table, as Supplementary Data 6, reporting fold-changes for viral loads in the different groups.

- Figure 3C-E: Sham groups for each first two panels (left and middle) seem to be identical and different from third (right). Please explain. It is somehow trivial that virus load in all vaccinated

animals differ significantly from sham (as indicated). A key question is however, how do infection outcomes differ between vaccine groups? Can this be implemented in the figures (or new table as discussed above).

We thank the reviewer for highlighting this question. Because of the large number of animals per cohort, we divided the challenge into two groups, the first one that compromised all animals primed with Ad26.COV2.S23 (n=99) and the second one the hamsters primed with BNT162b2 (n=50). For that reason, sham group data for each first two panels (left and middle) are identical (same sham-vaccinated animals) and different from third (right). We have included this explanation in Figure 3 legend (lines 275-276) and a summary table reporting fold-changes for viral loads in the different groups (Supplementary Data 6).

- Figure 4: I suggest to include a graphical presentation of histological scores (bar graph, spider plot, etc.) in the different groups for ease of comparison. Please include arrows or other symbols to highlight relevant pathologic changes (as mentioned in the results narrative: moderate to severe interstitial pneumonia characterized by hemorrhages, perivascularitis, perivascular edema, arterial endothelial leukocytic margination, peribronchiolitis, vasculitis and bronchiolitis).

We thank the reviewer for the suggestion, and we have added a bar graph with the histological scores as Supplementary Fig. 5. Since we were not able to include arrows or other symbols in Fig. 4, we added a column in the Supplementary Data 7 indicating exactly which section was chosen. In that way the readers can have the full histopathological evaluation from each section present in Fig. 4.

- Line 390: Statement many to be updated e.g. “BA5 and descendants such as BQ.1Q”
Thank you, we have updated the statement (lines 357-359).

- Line 408: “significant protection” to be defined. Neither of the used Covid-19 vaccines in any combination protected with high efficacy against infection and pathology in hamsters.

We have defined the “significant protection” as “significant reductions in BA.5 challenge viral loads” (line 376).

- Line 417-420: I agree that based on the immunogenicity and efficacy data presented herein, Ad26 is not inferior to other platforms such as mRNA. However, this statement regarding the Janssen vaccine may be a bit too strong. (i) The number of breakthrough cases in the 2x high Ad26 group is equally good (bad) as in 2x mRNA and worse than high Ad28 + mRNA (neglecting lack of statistical power due the small groups). (ii) Also Janssen uses a historical, mismatched antigen, whereas already updated (bivalent) mRNA vaccines are available. May the discussion hence not rather endorse an update to the promising yet old Ad26 vaccine?

We thank the reviewer for the discussion, we agree that the statement may be a bit too strong and we revised this statement (lines 388-395).

- Line 437: “increasing neutralizing antibody titers to up 3-months-post-booster and significantly decreasing viral loads in challenged infected hamsters”. This inverse correlation between nAb and viral loads is suggested in Figure 5. How strong do the authors judge the predictive power of nAb levels regarding protection against BA.5. From the data it appears more like “when nAb detectable than >90% chance to have lower viral load” thus more digital response. Are there clinical implications that could be drawn, also considering that data for 2 dpi (Figure S5) look much less promising as many more breakthrough infections can be observed?

We added a sentence (lines 413-415) highlighting that even high nAb titers do not completely prevent infection.

- Line 446: ... overestimation of protection ...

We have revised the expression and changed to “This could lead to an inability to compare protection offered against BA.5 compared to more virulent strains in this animal model” (line 422-423).

- Line 448: What is the possible skew in outcome the authors expect from using solely males? Regarding immune reaction, or pathology? Please give appropriate references.

We thank the reviewer for this comment. Our decision to use solely male Syrian hamsters were exclusively based on the cohort’s size. We chose to utilize relatively large cohort sizes to maintain power for the wide range of phenotypes measured. Due to the scale of this experiment, it was not feasible to double the animal numbers. As to determine sex-based differences, and to maintain the power required, the experimental and control groups would need to remain independent of each other for each sex in effect doubling the total amount of animals required. We do believe future, more focused studies should look at sex based differences.

- Line 453-459: The discussion of the doses used per vaccine should be more explicitly comparing doses used in previous studies in hamsters and in humans. E.g. what to conclude from the application of the 1/10 of a human dose in hamsters that weigh 1000x less than a humans. What is the evidence that these preclinical data translate into the clinics.

We thank the reviewer for highlighting this discussion. In reality, when expressed as a fractional human dose, all rodent studies have used very high doses. We selected our doses to be similar what has been used in previous hamster studies so that our data are comparable to other published works. We added this explanation to lines 436-438.

- Line 671: How were the lung sections scored and by whom (blinded/unblinded)?

The lung sections were analyzed blinded by Dr. Walker (Professor of Pathology at UTMB). The lung sections were first full described as showed in Supplementary Data 7. Visual observations of lung abnormalities in the left lung were scored from 0–4 depending on severity. The lung sections were analyzed for degree of involvement and scored as 0 (none, 0%), 1 (minimal, 1%–25%), 2 (mild, 26%–50%), 3 (moderate, 51%–75%), or 4 (severe, 76%–100%). We have included this information in the M&M section (lines 648-652).

- Line 950: Please mention in legend that presented Ab titers relate to prototypic strain.

We thank the reviewer for the suggestion, we have included this information (lines 836-840).

- Line 967: What is the relevance/meaning of the here mentioned 90% cutoff, like a euthanasia

We thank the reviewer for the question. We used the 90% weight cut-off (g), because our clinical scoring was based on >10% weight loss, as described in the M&M section (lines 546-552) and in the revised legend of Supplementary Data 4 (lines 856-868).

- Data given in extended data tables suffer from some lack of information in the legends (what are all the different values given in brackets?) and inconsistent use of “,” and “.” for numeric values.

We have reviewed all the extended data tables, included all relevant information in the legends and corrected the inconsistent use of “,” instead of “.” for numeric values.

Response to Reviewer #2 Comments

Reviewer #2 (Remarks to the Author):

Summary:

Machado et al. compare and contrast an experimental vaccine study in hamsters in which a series of homologous and heterologous prime (Pfizer/Janssen) / boost (Moderna, Novavax) vaccine regimens is compared, followed by challenge with BA.5. They track neutralizing antibody responses in the plasma of each cohort, as well as examining in vivo protection against weight loss, viral load, and lung histopathology.

Points of concern:

- The section spanning line 175-191, describing the boost neut data in Figure 2, is incredibly confusing and difficult to coordinate with the main figure.

We have revised this section (lines 148-165) for clarity and to better coordinate with Figure 2.

- o The NVX boost is discussed first but is found in the third column.

We have discussed first the NVX boost since it is our main data and we also kept in the third column because was the order standardized in the entire manuscript: PBS, mRNA-1273, NVX-CoV2373 and Ad26.COVS or BNT162b2. To avoid any misunderstood and better coordinate with the main text we have included the column indication after Fig.2 citation (lines 152-153).

- o In general, the paired analyses found in Fig. 2A-2C should likely be arranged in a group analyses similar to 2D-2F. It is difficult to track with the statements made in the main text.

We thank the reviewer for the suggestion; however it was not possible to arrange the data from 2a-c in a group analyses similar to 2d-f because the objective of 2a-c was to compare the nAb data within the same group in the different time-points. However, we agree with the reviewer that this may be confusing and we have revised the text regarding Fig. 2 to better explain it (lines 148-165).

- o Line 190: Should BNT162b2 actually be AD28-10⁸?

We thank the reviewer for the question. No, animals primed with BNT162b2 were not boosted with Ad26.COVS (10⁸ vp), only with NVX-CoV2373, mRNA-1273 and BNT162b2.

- o Line 187: The text goes between saying 3 months post-boost and day 252. This is confusing.

We thank the reviewer for highlighting this inconsistent nomenclature. Since day 252 represents 9 months post-primary vaccination and 3 months post-boost, we have standardized the text to 3 months post-boost (lines 160, 167, 320 and 337) and kept day 252 only when referring to 9 months post-vaccination (lines 80, 91, 96, 112, 114, 118, 121 and 140).

o Lines 193-202 I cannot understand. Without a figure to guide the analysis, attempting to interpret is very difficult.

We have revised the entire paragraph (lines 167-176) and shortened sentences to improve clarity. We have also included a summary table reporting key GMT plus 95% CI and the fraction of animals seroconverted for the different groups at selected timepoints for added clarity (Supplementary Data 3).

• Once again in Figure 3, the figures are incredibly difficult to analyze due to size/resolution and poor color contrast for data series; and the supplementary tables are formatted poorly to enable a reviewer to reasonably check. Figures HAVE to be more interpretable.

We thank the reviewer for highlighting the difficulty to analyze Figure 3 and supplementary data. We have changed color pallet from all the figures and increased size and resolution (600 dpi) for a better visualization of the data. The supplementary tables were also reviewed and kept in xlsx format to enable the readers to easily check the data.

o It appears the NVX boost is the winner in regards to challenge experiments. Lines 264-266 have this stated, but only refer to Fig. 3C. This should be Fig. 3C-3E.

We thank the reviewer for spotting this mistake and have corrected as suggested (line 238).

o It appears that the PBS boosts series for Ad26 prime data are quite similar to homologous/heterologous boosts in regards to weight loss. I don't think this was addressed in the main text and seems important in light of interpreting differences among the homologous/heterologous boosts.

We have addressed this observation by revising lines 229-232.

o Lines 274-277 address the above with the viral titer data. In general, this is a major point in regards to interpreting the boost data collectively. The figure (and main text) unfortunately seem to bury such findings in a mountain of other data.

We thank the reviewer for the comment. We have revised the section, made the proper corrections in order to highlight the viral titer data (236-250).

o Overall, this section of the main text poorly coordinates with the main figure. There is a lot of data here. Coordinating the results main text with an interpretable figure is a must and the bar is not meant here in that regard.

We thank the reviewer for the comment. We have revised the entire section, made the corrections suggested to better coordinated Figure 3 with the main text (206-260).

o Figure 3A challenge virus label appears to be incorrect

We thank the reviewer for spotting this typo and have corrected the label (Figure 3A).

• Section comprising lines 315-331, describing histology in Figure 4 seems to indicate conflicting descriptions of the histology. Lines 322-323 states "No apparent differences were observed among groups receiving different vaccines or boosters", yet some differences are described

throughout the section. However, I am not visually able to discern the described differences in the Figure 4 images, albeit I do not consider myself well trained in detected small differences in histopathology of lung tissue. With that said, if differences are meant to be readily discerned from these images, labels or a more concrete scoring method would be useful.

We thank the reviewer for highlight this point. We rewrote this sentence (lines 293-304) and included the scoring of the lung section for a better comparison (Supplementary Fig. 5 and Supplementary Data 7).

REVIEWERS' COMMENTS

Reviewer #1 (Remarks to the Author):

Many thanks to the authors for their thorough revision and great effort to improve their study, and for considering our suggestions. All my earlier concerns have been addressed appropriately.

Very few, minor remaining points are as follows:

1- In the discussion of the updated J&J vaccine (line 391-395), I suggest to delete the final expression "which still provide robust immune responses and protection against Omicron". Firstly, it may not be needed to state that updated Ad26.COVID.S.529 performs better against matched VOCs. Secondly I think such a statement poses the risk to overestimate respective vaccine potency against current VOC Omicron strains. In my humble view, vaccine-induced immune responses can only be defined as "robust" functionally, thus based on vaccine efficacy and when relevant. To the best of my knowledge (both in the clinics as well as in preclinical models) also Ad26.COVID.S was not able to consistently protect against SARS-CoV-2 VOC infection nor COVID-19 disease caused by VOCs, and definitely not from virus shedding/transmission. I would like to be a bit more hesitant to call any such vaccine "protective" without any further disclaimer. Clinical endpoints such as "protection from severe disease/mortality" or "marked reduction in lung pathology" may be minimal criteria, yet not ideal to suggest induction of particularly "robust" immunity.

2- Legends of Extended data files are not all self-explanatory

2a - Data 2: Lack of proper description of columns (What is reported? Reciprocal PRNT50 against Wuhan virus?)

2b - Data 3: typo in legend "Neutralizing"

2c - Data 6: Missing table title/no legend to clarify which numerical values are reported. We assume mean (or median ?) fold-changes of virus load, vRNA load etc. Please specify. Not clear whether "." used to separate "thousand" or "fractions" e.g. 16.339 = "less than 17" or "more than sixteen-thousand". Not clear how color code is used to score values, and how gradient from red to green is defined.

2d - Data 8: Please Specify abbreviations PPV, NPV and Kappa in legend

Reviewer #2 (Remarks to the Author):

See attached document

The authors addressed many concerns but below are the outstanding concerns that were not yet completely addressed.

o Line 190: Should BNT162b2 actually be AD28-10⁸?

We thank the reviewer for the question. No, animals primed with BNT162b2 were not boosted with Ad26.COVS.2 (10⁸ vp), only with NVX-CoV2373, mRNA-1273 and BNT162b2.

I see, then the Figure 2c legend is wrong. It should be BNT162b2 and not Ad26.COVS.2

o Lines 193-202 I cannot understand. Without a figure to guide the analysis, attempting to interpret is very difficult.

We have revised the entire paragraph (lines 167-176) and shortened sentences to improve clarity. We have also included a summary table reporting key GMT plus 95% CI and the fraction of animals seroconverted for the different groups at selected timepoints for added clarity (Supplementary Data 3).

This section still has what I believe to be either inconsistencies or has me confused without a figure to track the large amount of data presented. I will highlight some numbers that seem wrong according to your Supplementary Data 3 (with highlighted data). Either the data table is wrong or the text is wrong. Please double check your data or help me to understand what I am missing.

167 After 3 months post-boost, a higher proportion of animals boosted with NVX-CoV2373
 168 showed measurable FRNT50 titers against BA.5 (LOD=20); 54% (n=6/13) of the animals
 169 primed with 108 vp of Ad26.COVS.2, and sham-boosted, developed FRNT50 titers above
 170 the LOD. Hamsters boosted with mRNA-1273, NVX-CoV2373 and Ad26.COVS.2
 171 vaccines developed a measurable nAb response at rates of 50% (n=6/12), 25% (n=3/12)
 172 and 42% (n=5/12), respectively (Supplementary Data 3). Similar data were observed in
 173 the animals primed with 5 µg of BNT162b2, wherein no unboosted hamsters developed
 174 a measurable nAb response compared to 33% (n=4/12), 67% (n=8/12), and 25% (n=3/12)
 175 of those boosted with mRNA-1273, NVX-CoV2373, or BNT162b2, respectively
 176 (Supplementary Data 3).

Booster	Number of hamsters per time-point	Neutralizing antibodies (FRNT50) against BA.5 - Geometric mean titer (GMT) and 95%CI			Fraction and percentage (%) of animals seroconverted		
		Days post-vaccination (dpv)			Days post-vaccination (dpv)		
		168 (pre-boost)	196 (1mo post-boost)	251 (pre-challenge)	168 (pre-boost)	196 (1mo post-boost)	251 (pre-challenge)
PBS	59	10 (10-10)	10 (10-10)	10 (10-10)	N/A	N/A	N/A
PBS	14	36 (22-61)	33 (19-56)	31 (18-53)	12/14 (85.7)	12/14 (85.7)	12/14 (85.7)
mRNA-1273 (0.5µg)	12	28 (13-59)	42 (24-75)	40 (22-72)	8/12 (66.7)	11/12 (91.7)	11/12 (91.7)
NVX-CoV2373 (1µg rS/15ug Matrix-M)	12	30 (12-73)	80 (33-193)	63 (23-172)	6/12 (50.0)	10/12 (83.3)	10/12 (83.3)
Ad26.COVS.2 (10 ⁸ 8vp)	12	25 (13-49)	30 (14-63)	25 (13-50)	8/12 (66.7)	8/12 (66.7)	7/12 (58.3)
PBS	13	25 (13-47)	26 (14-50)	22 (12-40)	8/13 (61.5)	8/13 (61.5)	7/13 (53.8)
mRNA-1273 (0.5µg)	12	15 (8-28)	25 (11-57)	25 (11-55)	3/12 (25.0)	6/12 (50.0)	6/12 (50.0)
NVX-CoV2373 (1µg rS/15ug Matrix-M)	12	13 (9-15)	50 (23-110)	50 (22-115)	2/12 (16.7)	9/12 (75.0)	9/12 (75.0)
Ad26.COVS.2 (10 ⁸ 8vp)	12	20 (8-48)	30 (12-74)	28 (12-69)	3/12 (25.0)	7/12 (58.3)	7/12 (58.3)
PBS	14	10 (10-10)	10 (9-12)	10 (10-10)	0/14 (0.0)	1/14 (7.1)	0/14 (0.0)
mRNA-1273 (0.5µg)	12	11 (9-13)	16 (10-26)	15 (9-24)	2/12 (16.7)	4/12 (33.3)	3/12 (25.0)
NVX-CoV2373 (1µg rS/15ug Matrix-M)	12	11 (9-12)	32 (17-58)	34 (17-66)	1/12 (8.3)	11/12 (91.7)	8/12 (66.7)
Bnt162b2 (0.5µg)	12	11 (9-13)	15 (10-22)	11 (9-13)	1/12 (8.3)	4/12 (33.3)	2/12 (16.7)

Response to Reviewer #1 Comments

Reviewer #1 (Remarks to the Author):

Many thanks to the authors for their thorough revision and great effort to improve their study, and for considering our suggestions. All my earlier concerns have been addressed appropriately.

Very few, minor remaining points are as follows:

1- In the discussion of the updated J&J vaccine (line 391-395), I suggest to delete the final expression “which still provide robust immune responses and protection against Omicron”. Firstly, it may not be needed to state that updated Ad26.COVS.529 performs better against matched VOCs. Secondly I think such a statement poses the risk to overestimate respective vaccine potency against current VOC Omicron strains. In my humble view, vaccine-induced immune responses can only be defined as “robust” functionally, thus based on vaccine efficacy and when relevant. To the best of my knowledge (both in the clinics as well as in preclinical models) also Ad26.COVS was not able to consistently protect against SARS-CoV-2 VOC infection nor COVID-19 disease caused by VOCs, and definitely not from virus shedding/transmission. I would like to be a bit more hesitant to call any such vaccine “protective” without any further disclaimer. Clinical endpoints such as “protection from severe disease/mortality” or “marked reduction in lung pathology” may be minimal criteria, yet not ideal to suggest induction of particularly "robust" immunity.

We thank the reviewer for the discussion and suggestion. We have deleted the statement “which still provide robust immune responses and protection against Omicron”, as requested. We agree with your considerations and changed the “protective” expression to “showed a marked reduction in lung viral loads and pathology” (lines 402-408).

2- Legends of Extended data files are not all self-explanatory
2a - Data 2: Lack of proper description of columns (What is reported? Reciprocal PRNT50 against Wuhan virus?)

We have included a description of the data as “Neutralizing antibodies (FRNT50) titers against BA.5” on the top of columns and clarified the legend: Supplementary Data 2. Neutralizing antibody titers expressed as FRNT50 values, pre-, 1mo post- and 3mo post-boost against Omicron BA.5 sublineage. Titters below the LOD (<20) were represented by half the LOD, 10. ID, identification; mo, month(s); dpv, days post-vaccination; vp, viral particles; µg, micrograms; FRNT50, 50% focus reduction neutralization antibody titer (lines 861-865).

2b - Data 3: typo in legend “Neutralizing”

We thank the reviewer for spotting this typo and have corrected as requested (Supplementary Data 3).

2c - Data 6: Missing table title/no legend to clarify which numerical values are reported. We assume mean (or median ?) fold-changes of virus load, vRNA load etc. Please specify. Not clear whether “.” used to separate “thousand” or “fractions” e.g. 16.339 = “less than 17” or “more than sixteen-thousand”. Not clear how color code is used to score values, and how gradient from red to green is defined.

Thank you for your comment, we addressed this by adding all information requested in the legend: “Supplementary Data 6. Fold change reduction of infectious virus titer (virus load) in nasal washes, trachea, and lung 2- and 4-days post-infection (dpi) with Omicron BA.5. The fold change reduction was calculated as the ratio of the mean values. Sham-vaccinated group was used for comparison and fold change reduction calculation. Comma (,) were used as thousands separator. Numbers are color scaled for greater visualization from the lowest to the highest values (green → white → red). For a comparable data visualization, the color scale was applied separately in each specimen at each time-point from each primary vaccinated group” (lines 888-895).

2d - Data 8: Please Specify abbreviations PPV, NPV and Kappa in legend

We thank you the reviewer for the suggestion. We have specified the abbreviations PPV, NPV and Kappa in the main data as in the legend: Supplementary Data 8. Analysis of agreement between neutralizing titer (FRNT50) and infectious virus titer in lungs at 4 days post-infection. Sensitivity, specificity, positive and negative predictive values (PPV and NPV, respectively) and Cohen's Kappa coefficient (value of kappa) are shown, with 95% confidence interval (CI) (lines 901-904).

Response to Reviewer #2 Comments

The authors addressed many concerns but below are the outstanding concerns that were not yet completely addressed.

o Line 190: Should BNT162b2 actually be AD28-10⁸?

We thank the reviewer for the question. No, to clarify for the reviewer, animals primed with BNT162b2 were not boosted with Ad26.COVS (10⁸ vp), only with NVX-CoV2373, mRNA-1273 and BNT162b2.

I see, then the Figure 2c legend is wrong. It should be BNT162b2 and not Ad26.COVS

We thank the reviewer for spotting this typo and have made the correction, as requested (Figure 2c).

o Lines 193-202 I cannot understand. Without a figure to guide the analysis, attempting to interpret is very difficult.

We have revised the entire paragraph (lines 167-176) and shortened sentences to improve clarity. We have also included a summary table reporting key GMT plus 95% CI and the fraction of animals seroconverted for the different groups at selected timepoints for added clarity (Supplementary Data 3).

This section still has what I believe to be either inconsistencies or has me confused without a figure to track the large amount of data presented. I will highlight some numbers that seem wrong according to your Supplementary Data 3 (with highlighted data). Either the data table is wrong or the text is wrong. Please double check your data or help me to understand what I am missing.

167 After 3 months post-boost, a higher proportion of animals boosted with NVX-CoV2373
168 showed measurable FRNT50 titers against BA.5 (LOD=20); 54% (n=6/13) of the animals
169 primed with 10⁸ vp of Ad26.COVS, and sham-boosted, developed FRNT50 titers above
170 the LOD. Hamsters boosted with mRNA-1273, NVX-CoV2373 and Ad26.COVS
171 vaccines developed a measurable nAb response at rates of 50% (n=6/12), 25% (n=3/12)
172 and 42% (n=5/12), respectively (Supplementary Data 3). Similar data were observed in
173 the animals primed with 5 × 10⁸ of BNT162b2, wherein no unboosted hamsters developed
174 a measurable nAb response compared to 33% (n=4/12), 67% (n=8/12), and 25% (n=3/12)
175 of those boosted with mRNA-1273, NVX-CoV2373, or BNT162b2, respectively
176 (Supplementary Data 3).

We thank the reviewer for highlighting this issue. We have reviewed the entire data, made all proper corrections, and updated the numbers in the main text (lines 179-189 and Supplementary Data 3). Though we changed the text in the aforementioned section to meet your recommendations, we wanted to add a level of clarity due to the complexity of what was presented. As such we added a minor supplemental figure (Supplementary Fig. 4) to add a visual representation of the data presented per your request.